# PRM: PHOTOMETRIC STEREO BASED LARGE RECONSTRUCTION MODEL

## ABSTRACT

We propose PRM, a novel photometric stereo based large reconstruction model to reconstruct high-quality meshes with fine-grained local details. Unlike previous large reconstruction models that prepare images under fixed and simple lighting as both input and supervision, PRM renders photometric stereo images by varying materials and lighting for the purposes, which not only improves the precise local details by providing rich photometric cues but also increases the model's robustness to variations in the appearance of input images. To offer enhanced flexibility of images rendering, we incorporate a real-time physically-based rendering (PBR) method and mesh rasterization for online images rendering. Moreover, in employing an explicit mesh as our 3D representation, PRM ensures the application of differentiable PBR, which supports the utilization of multiple photometric supervisions and better models the specular color for high-quality geometry optimization. Our PRM leverages photometric stereo images to achieve high-quality reconstructions with fine-grained local details, even amidst sophisticated image appearances. Extensive experiments demonstrate that PRM significantly outperforms other models.

## 1 INTRODUCTION

Recent advancements in generative models (Song et al., 2020; Ho et al., 2020) have spurred notable progress in 2D content creation, driven by fast growth in data volumes. In contrast, the development in 3D field remains encumbered due to limited 3D assets, which are essential for diverse applications including game modeling (Gregory, 2018), computer animation (Parent, 2012; Lasseter, 1987), and virtual reality (Schuemie et al., 2001). Traditional approaches to generating 3D assets have utilized optimization-based techniques from multi-view posed images (Wang et al., 2021; Yariv et al., 2021; 2020) or have harnessed SDS-based distillation methods from 2D diffusion models (Liang et al., 2023; Lin et al., 2023; Poole et al., 2022). Despite their effectiveness, these methods often require increased computational costs without a commensurate improvement in surface quality, making them less suitable for rapid deployment in real-world scenarios.

Feed-forward 3D generative models (Hong et al., 2023; Hu et al., 2024; Zhang et al., 2024) have been developed to address the limitations of per-scene optimization by training a generalizable model on large-scale 3D assets. Notably, the Large Reconstruction Model (LRM) (Hong et al., 2023) has demonstrated promising results, exhibiting exceptional reconstruction speeds. The subsequent LRM series (Hong et al., 2023; Xu et al., 2023; Wang et al., 2024; Xu et al., 2024; Tang et al., 2024) utilizes a Transformer-based architecture to encode either single or multi-view images, and decoding them into 3D representations, such as triplanes (Chan et al., 2022), Flexicubes (Shen et al., 2023) or 3D Gaussians (Tang et al., 2024). These 3D representations enable differentiable rendering from arbitrary viewpoints, which is crucial for calculating multi-view reconstruction loss for optimization.

While LRM series demonstrate effectiveness and efficiency in globally coherent 3D assets reconstruction, they encounter limitations in accurately capturing fine-grained local details. This challenge stems from their dependence on images rendered under fixed and simple lighting conditions, which provide inadequate photometric information for detailed surface reconstruction. Furthermore, LRM series are sensitive to variations in the appearance of conditioned images, particularly when dealing with surfaces exhibiting glossy characteristics. As a result, LRM series tend to entangle the texture and geometry, leading to wrong geometry reconstruction.

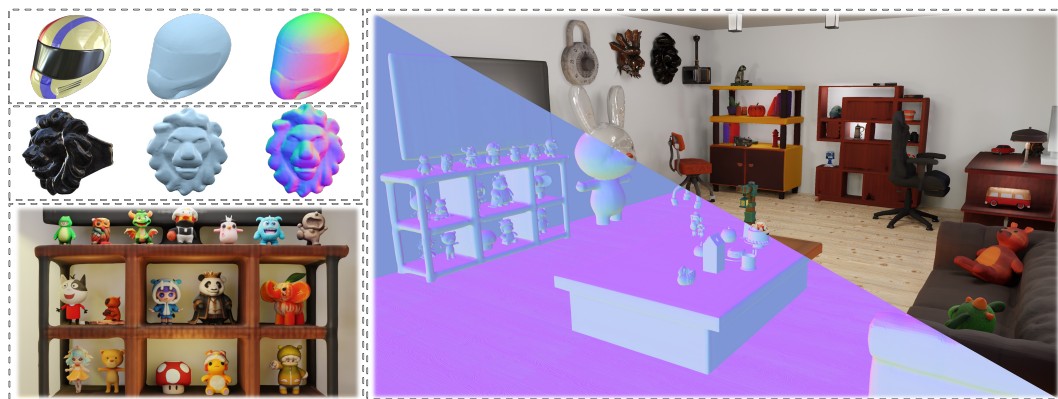

Figure 1: Top left: PRM is capable of reconstructing high-quality meshes with fine-grained local details even under complex image appearances, such as specular highlights and dark appearances. Right: We demonstrate a scene comprising diverse 3D objects generated by our models. Bottom left: A zoomed-in visualization of the scene highlights these details more clearly.

To address the above mentioned challenges, we introduce PRM, a photometric stereo based large reconstruction model. This model is adept at capturing fine-grained local details and ensures robustness against the complex appearances of input images. We achieve these objectives by leveraging photometric stereo images (Hernandez et al., 2008). Specifically, we render photometric stereo images by varying camera pose, materials (i.e., metallic and roughness), and lighting for both input and supervision. However, rendering these images is not trivial since there are infinite possible combinations of camera pose, materials and lighting. In the recent LRM series, images are typically rendered offline using Blender's Cycles engine (Hess, 2013). While this approach produces high-quality, noise-free images, it requires numerous samples of lighting directions, significantly increasing the time cost and making it expensive to maximize the training sample distribution.

To address this issue, we incorporate a real-time, physically based rendering technique known as split-sum approximation (Karis & Games, 2013), along with mesh rasterization for online rendering. This approach offers greater flexibility compared to traditional offline methods. We discuss two corresponding training strategies in the Appendix, thanks to the flexible rendering. Photometric stereo images offer two distinct advantages. First, photometric stereo images furnish additional photometric cues, thereby enhancing the capacity to recover fine-grained local details. Second, the PRM model demonstrates remarkable robustness to variations in the appearances of input images. For instance, it is capable of accurately reconstructing the geometry of images with glossy surfaces. Moreover, by utilizing mesh as our 3D representation, we are capable of utilizing differentiable PBR to produce intermediate shading variables such as albedo, specular light, and diffuse light maps, along with geometric cues like normals and depth. These variables provide multiple supervisions, including photometric supervision and geometric supervision for high-quality geometry reconstruction. Furthermore, PBR can better disentangle the specular component, making the geometry also be correctly recovered when the supervision images are characterized with glossy surfaces.

To summarize, our contributions are listed as follows.

- We introduce PRM, a model that is capable of reconstructing geometry with fine-grained local details and robust to variations in the appearance of input image by utilizing photometric stereo images as input and supervision.

- To the best of our knowledge, we are the first to integrate split-sum approximation and mesh rasterization to render images online for LRM, offering significantly greater flexibility.

- By utilizing mesh as the 3D representation, we ensure differentiable PBR for predictive rendering. This approach is advantageous for modeling reflective components and enables the incorporation of multiple supervisions for high-quality geometry reconstruction, significantly outperforming other models.

## 2 RELATED WORK

### 2.1 FEED-FORWARD 3D GENERATIVE MODELS

Large-scale 3D assets (Deitke et al., 2023) facilitate the training of generalizable reconstruction models. Recent works have focused on generating 3D objects using feed-forward models (Hong et al., 2023; Xu et al., 2023; Wang et al., 2024; Xu et al., 2024; Li et al., 2023; Hu et al., 2024; Tang et al., 2024; Zhang et al., 2024), demonstrating impressive results in terms of speed and quality. Specifically, Clay (Zhang et al., 2024) utilizes occupancy for direct supervision. X-ray (Hu et al., 2024) explores novel 3D representations by converting a 3D object into a series of surface frames at different layers. The LRM series (Hong et al., 2023; Xu et al., 2023; Wang et al., 2024; Xu et al., 2024; Li et al., 2023) shows that a transformer backbone can effectively map image tokens to 3D triplanes, benefiting from multi-view supervision. Instant3D (Li et al., 2023) employs multi-view images to provide additional 3D information for triplane prediction, yielding promising outcomes. CRM (Wang et al., 2024) and InstantMesh (Xu et al., 2024) opt for an explicit mesh representation, supporting mesh rasterization and rendering additional geometric cues for supervision. Despite these achievements, the existing LRM series typically render low-frequency images under fixed and simple lighting, which compromises the model's adaptability to complexity in the appearance of input images and the model's capability to recover local details due to limited photometric cues. In response, we render photometric stereo images that significantly enhance the photometric cues necessary for the recovery of fine-grained local details.

### 2.2 PHOTOMETRIC STEREO

Photometric stereo (PS) is a technique for recovering surface normals from the appearance of an object under varying lighting conditions. Traditional methods, inspired by the seminal work (Woodham, 1980), assume calibrated, directional lighting. Recently, uncalibrated photometric stereo methods have emerged, which assume Lambertian integrable surfaces and aim to resolve the General Bas-Relief ambiguity (Hayakawa, 1994) between light and geometry. However, these methods are still constrained to single directional lighting. More contemporary research (Mo et al., 2018; Ikehata, 2022; 2023) has shifted focus towards natural lighting conditions. Despite the significant progress, these approaches generally concentrate on single-view photometric stereo, relying solely on photometric cues and neglecting multi-view information, which is crucial for accurately reasoning geometric features. Some studies (Kaya et al., 2022a; Park et al., 2013; Zhao et al., 2023; Hernandez et al., 2008; Kaya et al., 2022b; 2023) leverage both photometric and geometric cues for reconstruction. These cues are complementary: photometric stereo provides precise local details, while multi-view information yields accurate global shapes (Zhao et al., 2023). For example, UA-MVPS (Kaya et al., 2022a) utilizes complementary strengths of PS and multi-view stereo for geometry reconstruction. NeRF-MVPS (Kaya et al., 2022b) utilizes surface normal estimated from photometric stereo images to enhance the reconstruction performance of NeRF. Our approach integrates the principles of photometric stereo into LRM, aiming to harness the strengths of photometric cues for enhanced reconstruction accuracy.

### 2.3 PHYSICALLY-BASED RENDERING

Physically based rendering (PBR) is a computer graphics approach that renders photo-realistic images. PBR offers a physically plausible approach to modeling radiance by simulating the interaction between lighting and materials. PBR has proven to be effective in improving the geometry for multi-view reconstruction task. For example, incorporating the principles of PBR into volume rendering significantly improves the accuracy (Verbin et al., 2022; Ge et al., 2023; Liu et al., 2023), especially for glossy surfaces. Since geometry and predicted radiance are closely entangled, improving radiance modeling can also enhance geometry reconstruction. Besides, PBR is also widely used in inverse rendering task (Barron & Malik, 2014; Nimier-David et al., 2019). The task aims at decomposing image appearance into intrinsic properties. Unlike previous LRM methods that predict radiance without explicitly considering the interactions between materials and lighting, we leverage advancements from the multi-view reconstruction field and employ PBR for improved radiance modeling and geometry reconstruction. To this end, we predict albedo instead of color, which is more reasonable as albedo is view-independent. The final color is derived using the predicted albedo and the ground truth metallic, roughness, and lighting.

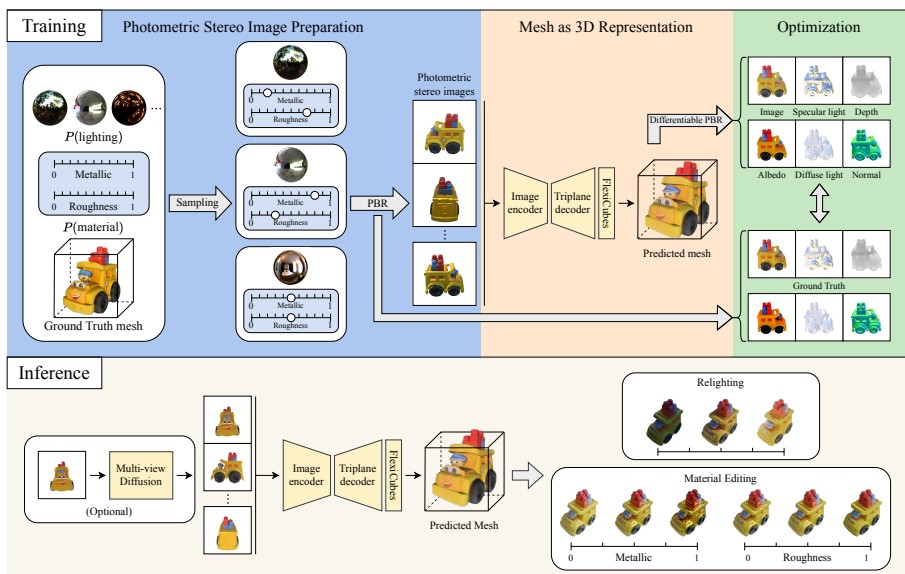

Figure 2: Overview of our framework. During training, photometric stereo images are rendered using PBR with randomly varied materials, lighting, and camera poses, along with depth, normal, albedo, and lighting maps. Images are encoded as a mesh through the network. All associated maps, along with the images, are used for supervision. During inference, an optional multi-view diffusion model takes a single image as input and outputs multi-view images, which are then fed into the network for mesh prediction. Relighting and material editing functionalities are also supported.

## 3 METHOD

We begin with a succinct overview of large reconstruction model, physically based rendering and photometric stereo in Section 3.1. Then, we introduce how to prepare photometric stereo images in Section 3.2. Subsequently, we introduce PRM in Section 3.3, with our proposed comprehensive objectives and applications. An overview of our framework is provided in Figure 2.

### 3.1 PRELIMINARIES

**Large Reconstruction Model** aims to reconstrcut 3D assets given a single image or multi-view images. LRM first utilizes a pre-trained visual transformer, DINO (Caron et al., 2021), to encode the images into image tokens. Subsequently, it employs an image-to-triplane transformer decoder that projects these 2D image tokens onto a 3D triplane using cross-attention (Hong et al., 2023). Following this, images can be differentiable rendered from any viewpoint by decoding the triplane features into color and density, supporting photometric supervision and optimization.

**Physically-based Rendering** aims to produce 2D images using specified geometry, materials, and lighting. Central to this process is the rendering equation (Kajiya, 1986) formulated by

$$\boldsymbol{C}(\boldsymbol{x}, \boldsymbol{\omega_o}) = \int_\Omega f(\boldsymbol{x}, \boldsymbol{\omega_o}, \boldsymbol{\omega_i}) L_i(\boldsymbol{x}, \boldsymbol{\omega_i})(\boldsymbol{\omega_i} \cdot \mathbf{n}) d\boldsymbol{\omega_i}, \tag{1}$$

where $\boldsymbol{\omega_o}$ is the viewing direction of the outgoing light, $L_i$ is the incident light of direction $\boldsymbol{\omega_i}$ sampled from the upper hemisphere $\Omega$ of the surface point $\boldsymbol{x}$, and $\mathbf{n}$ is the surface normal. $f$ is the BRDF properties. The function $f$ consists of a diffused and a specular component

$$f(\boldsymbol{x}, \boldsymbol{\omega_o}, \boldsymbol{\omega_i}) = (1 - m)\frac{\boldsymbol{a}}{\pi} + \frac{DFG}{4(\boldsymbol{\omega_i} \times \mathbf{n})(\boldsymbol{\omega_o} \times \mathbf{n})}, \tag{2}$$

where $m \in [0, 1]$ is the metallic, $\boldsymbol{a} \in [0, 1]^3$ is the albedo. We detail the expression of $D$, $F$ and $G$ in the Appendix. With Eq.(1) and Eq.(2), the outgoing radiance is given by

$$\boldsymbol{C}(\boldsymbol{x}, \boldsymbol{\omega_o}) = \boldsymbol{C}_{\mathrm{d}}(\boldsymbol{x}, \boldsymbol{\omega_o}) + \boldsymbol{C}_{\mathrm{s}}(\boldsymbol{x}, \boldsymbol{\omega_o}), \tag{3}$$

$$C_\mathrm{d}(\boldsymbol{x}, \boldsymbol{\omega}_o) = (1-m)\boldsymbol{a}\int_\Omega L_i(\boldsymbol{x}, \boldsymbol{\omega_i})\frac{(\boldsymbol{\omega_i} \cdot \mathbf{n})}{\pi}d\boldsymbol{\omega_i}, \tag{4}$$

$$C_\mathrm{s}(\boldsymbol{x}, \boldsymbol{\omega_o}) = \int_\Omega \frac{DFG}{4(\boldsymbol{\omega_i} \times \mathbf{n})(\boldsymbol{\omega_o} \times \mathbf{n})}L_i(\boldsymbol{x}, \boldsymbol{\omega_i})(\boldsymbol{\omega_i} \cdot \mathbf{n})d\boldsymbol{\omega_i}, \tag{5}$$

$C_\mathrm{s}$ and $C_\mathrm{d}$ are specular and diffuse color, respectively.

**Photometric Stereo** aims to estimate the surface normals by observing an object under varying lightings (Woodham, 1980). When considering a Lambertian surface illuminated by a single point-like, distant light, the color is determined solely by the diffuse term, which can be formulated by

$$C(\boldsymbol{x}) = \boldsymbol{a}(\boldsymbol{L} \cdot \boldsymbol{n}), \tag{6}$$

where $\boldsymbol{L} = L \cdot \boldsymbol{\omega}$, $\boldsymbol{\omega}$ and $L$ are the direction and the intensity of the lighting, respectively. When we observe the object under different lighting conditions $\boldsymbol{L}_i$, we have multiple such equations. We assume that both the direction and intensity of the lighting are known, and the shading colors $C(\boldsymbol{x})$ is also observed. Under these conditions, determining the surface normals $\boldsymbol{n}$ and the albedo $\boldsymbol{a}$ is effectively equivalent to solving a system of equations derived from these observations. More lighting indicates more equations, which effectively constrain the solution space.

### 3.2 Photometric Stereo Images Preparation

Previous methods typically prepared training data by rendering multi-view images using fixed, simple lighting and materials in Blender (Hess, 2013). This approach resulted in images characterized by low-frequency appearances, providing limited photometric cues. Consequently, these methods struggled to reconstruct geometry with precise local details. Moreover, they often fail when processing images with glossy surfaces, as the models tend to interpret these glossy attributes as geometric permutations. An example is shown in Figure 3. Our method correctly reconstructs surfaces with glossy component.



Input image     InstantMesh     Ours

Figure 3: Comparison on shiny objects.

In contrast, we prepare photometric stereo images by varying materials and lighting. A naive solution is to prepare these images offline, as with previous methods, but this approach poses significant challenges due to the infinite number of potential combinations of materials and lighting. Moreover, rendering high-quality images requires large sample counts, making traditional data preparation methods infeasible. To overcome these issues, we incorporate a real-time rendering method known as split-sum approximation (Karis & Games, 2013) along with mesh rasterization, which facilitates rapid rendering. This method enables online data preparation and significantly enhances flexibility.

**Split-sum approximation.** High-quality estimation of physically based rendering typically requires Monte Carlo sampling to approximate the integral in Eq.(1). However, this process demands large sample counts, making it time-consuming. Instead, we employ a real-time rendering method known as the split-sum approximation (Karis & Games, 2013). According to the split-sum approximation, the specular component in Eq.(5) can be rewritten as:

$$C_\mathrm{s}(\boldsymbol{x}, \boldsymbol{\omega}_o) \approx \int_\Omega \frac{DFG}{4(\boldsymbol{\omega}_o \cdot \boldsymbol{n})}d\boldsymbol{\omega}_i \int_\Omega L(\boldsymbol{x}, \boldsymbol{\omega}_i)D(\hat{\boldsymbol{d}}, \rho)d\boldsymbol{\omega}_i, \tag{7}$$

The first term is the integral of the BRDF, which is approximated by specular albedo $\boldsymbol{a}_\mathrm{s} = ((1-m)*0.04+m*\boldsymbol{a})*F_1+F_2$, where $F_1$ and $F_2$ are pre-computed scalars and stored in a 2D lookup texture related to $\rho, \boldsymbol{n}$ and $\boldsymbol{\omega}_o$. The second term is the integral of lights on the normal distribution function $D(\hat{\boldsymbol{d}}, \rho)$, which can also be pre-computed and stored as mipmaps $\boldsymbol{M}_\mathrm{s}$. $\hat{\boldsymbol{d}} = 2(-\boldsymbol{\omega}_o \cdot \boldsymbol{n})\boldsymbol{n} + \boldsymbol{\omega}_o$ is the reflective direction. After simplification, the Eq.(7) is modified as

$$C_\mathrm{s}(\boldsymbol{x}, \boldsymbol{\omega}_o) = \boldsymbol{a}_\mathrm{s}(\boldsymbol{a}, m, \boldsymbol{n}, \boldsymbol{\omega}_o)\boldsymbol{L}_\mathrm{spec}(\boldsymbol{x}, \boldsymbol{n}, \boldsymbol{\omega}_o, \rho, \boldsymbol{M}_\mathrm{s}), \tag{8}$$

where $\boldsymbol{L}_\mathrm{spec} = \mathrm{Tex\_Sample}(\boldsymbol{x}, \hat{\boldsymbol{d}}, \rho, \boldsymbol{M}_\mathrm{s})$, $\mathrm{Tex\_Sample}$ indicates texture sampling based on different levels of roughness $\rho$ in pre-computed lighting map $\boldsymbol{M}_\mathrm{s}$. A low-resolution map $\boldsymbol{M}_\mathrm{d}$ is also created to represent low-frequency diffuse lighting and the diffuse part in Eq.(4) is simplified as

$$C_\mathrm{d}(\boldsymbol{x}, \boldsymbol{\omega}_o) = \boldsymbol{a}_\mathrm{d}(\boldsymbol{a}, m)\boldsymbol{L}_\mathrm{diff}(\boldsymbol{x}, \boldsymbol{n}, \boldsymbol{M}_\mathrm{d}), \tag{9}$$

where $\boldsymbol{a}_{\mathrm{d}} = (1 - m)\boldsymbol{a}$ indicates the diffuse albedo and $\boldsymbol{L}_{\mathrm{diff}} = \mathrm{Tex\_Sample}(\boldsymbol{x}, \boldsymbol{n}, \boldsymbol{M}_{\mathrm{d}})$. We show some rendered examples in the Appendix in Figure 10.

**Discussion.** We render photometric stereo images using varied camera poses, materials, and lighting conditions rather than solely changing the lighting. Please refer to the Appendix for more details. This approach offers two distinct advantages. Firstly, multi-view images provide more geometric cues than single-view images, which are crucial for reconstructing globally reasonable geometry (Kaya et al., 2022b;a; 2023). Secondly, by varying materials, we can create images with glossy appearances, particularly when the metallic component is high and roughness is low. These varied images serve as inputs, enhancing the model's robustness to variations in appearance. Furthermore, the core principles of photometric stereo remain applicable. In equations Eq.(8) and Eq.(9), the observation direction $\boldsymbol{\omega}_o$, metallic value $m$, roughness $\rho$, and mipmaps $\boldsymbol{M}_{\mathrm{d}}$ and $\boldsymbol{M}_{\mathrm{s}}$ are all known. Predicting the surface normals $\boldsymbol{n}$ and the albedo $\boldsymbol{a}$ still equates to solving these equations. Moreover, compared to merely changing lighting, altering the metallic and roughness values allows for diverse shading color rendering, which produces a richer set of equations.

**Mesh Rasterization Rendering.** Given an object with explicit mesh $\boldsymbol{O}$, rasterization is utilized to determine surface points $\boldsymbol{x}$, along with corresponding depth $\boldsymbol{d}$, surface normals $\boldsymbol{n}$, and mask $\boldsymbol{m}$. After obtaining the surface points $\boldsymbol{x}$ and their surface normals $\boldsymbol{n}$ along with selected camera pose, materials and lighting, we leverage split-sum approximation to estimate the specular and diffuse color as Eq.(8) and Eq.(9), respectively. During the process, besides the shading color, we can also render albedo, specular light, and diffuse light maps. The entire process can be formulated as

$$\{\boldsymbol{C}, \boldsymbol{n}, \boldsymbol{d}, \boldsymbol{m}, \boldsymbol{a}, \boldsymbol{L}_{\mathrm{spec}}, \boldsymbol{L}_{\mathrm{diff}}\} = \mathrm{PBR}(\mathrm{Rasterization}(\boldsymbol{O})), \tag{10}$$

where $\boldsymbol{C}, \boldsymbol{n}, \boldsymbol{d}, \boldsymbol{m}, \boldsymbol{a}, \boldsymbol{L}_{\mathrm{spec}}$, and $\boldsymbol{L}_{\mathrm{diff}}$ are the rendered color, normal, depth, mask, albedo, specular light, and diffuse light maps, respectively. We show some rendered cases in the Appendix.

### 3.3 PRM

**Mesh as 3D Representation.** The previous LRM-based models typically integrate triplane as 3D representation. In contrast, we opt for an explicit representation using mesh as our 3D format, which enables the use of the same PBR method employed in data preparation. As a result, specular and diffuse lighting maps are also renderable, providing extra photometric cues that are only related to surface normals. Moreover, PBR can effectively model the specular component, leading to improved geometry reconstruction results (Verbin et al., 2022; Ge et al., 2023). Specifically, we leverage differentiable iso-surface extraction module, namely FlexiCubes (Shen et al., 2023).

**Two Stage Optimization.** Inspired by InstantMesh (Xu et al., 2024), we have similarly designed a two-stage optimization framework. The first stage mirrors Instantmesh, using triplane and volume rendering for optimization with offline rendered data. In the second stage, FlexiCubes is used as the 3D representation. To reuse the knowledge in the first stage, we load the pretrained model as in InstantMesh (Xu et al., 2024). The original color MLP is repurposed as an albedo MLP to utilize the color priors. Since an explicit mesh is utilized as our 3D representation, we can render novel views as described in Eq.(10). The difference is that the mesh $\hat{\boldsymbol{O}}$ is extracted using the dual marching cubes algorithm (Nielson, 2004), which utilizes predicted SDF values, deformation, and weights derived from the triplane formulated by

$$\{\hat{\boldsymbol{C}}, \hat{\boldsymbol{n}}, \hat{\boldsymbol{d}}, \hat{\boldsymbol{m}}, \hat{\boldsymbol{a}}, \hat{\boldsymbol{L}}_{\mathrm{spec}}, \hat{\boldsymbol{L}}_{\mathrm{diff}}\} = \mathrm{PBR}(\mathrm{Rasterization}(\hat{\boldsymbol{O}})). \tag{11}$$

**Optimization.** During the training process, our total loss function is

$$\mathcal{L} = \mathcal{L}_{\mathrm{MSE}}(\boldsymbol{C}, \hat{\boldsymbol{C}}) + \lambda_{\mathrm{LPIPS}}\mathcal{L}_{\mathrm{LPIPS}}(\boldsymbol{C}, \hat{\boldsymbol{C}}) + \mathcal{L}_{\mathrm{MSE}}(\boldsymbol{a}, \hat{\boldsymbol{a}}) + \lambda_{\mathrm{LPIPS}}\mathcal{L}_{\mathrm{LPIPS}}(\boldsymbol{a}, \hat{\boldsymbol{a}})$$
$$+ \mathcal{L}_{\mathrm{MSE}}(\boldsymbol{L}_*, \hat{\boldsymbol{L}}_*) + \lambda_{\mathrm{LPIPS}}\mathcal{L}_{\mathrm{LPIPS}}(\boldsymbol{L}_*, \hat{\boldsymbol{L}}_*) + \lambda_{\mathrm{normal}}\hat{\boldsymbol{m}} \otimes (1 - \boldsymbol{n} \cdot \hat{\boldsymbol{n}}) + \lambda_{\mathrm{reg}}\mathcal{L}_{\mathrm{reg}} \tag{12}$$
$$+ \lambda_{\mathrm{depth}}\hat{\boldsymbol{m}} \otimes \|\boldsymbol{d} - \hat{\boldsymbol{d}}\|_1 + \lambda_{\mathrm{mask}}\mathcal{L}_{\mathrm{MSE}}(\boldsymbol{m}, \hat{\boldsymbol{m}}),$$

where $\mathcal{L}_{\mathrm{MSE}}$ and $\mathcal{L}_{\mathrm{LPIPS}}$ indicates the mean squaree error loss and LPISP loss (Zhang et al., 2018), respectively. $* \in \{\mathrm{spec}, \mathrm{diff}\}$ denotes specular light and diffuse light maps, respectively. $\mathcal{L}_{\mathrm{reg}}$ is the regularization terms used in FlexiCubes (Shen et al., 2023). During training, we set $\lambda_{\mathrm{LPIPS}} = 2.0$, $\lambda_{\mathrm{normal}} = 0.2$, $\lambda_{\mathrm{depth}} = 0.5$, $\lambda_{\mathrm{mask}} = 1.0$ and $\lambda_{\mathrm{reg}} = 0.01$.

Table 1: Quantitative comparison with state-of-the-art methods on the GSO and Omni3D datasets, showcasing 3D reconstruction and 2D rendering metrics.

| Dataset | GSO | | | | | OmniObject3D | | | | |
|---|---|---|---|---|---|---|---|---|---|---|
| Metric | CD↓ | FS@0.1↑ | PSNR↑ | SSIM↑ | LPIPS↓ | CD↓ | FS@0.1↑ | PSNR↑ | SSIM↑ | LPIPS↓ |
| TripoSR | 0.109 | 0.872 | 15.247 | 0.859 | 0.197 | 0.083 | 0.940 | 15.237 | 0.865 | 0.176 |
| CRM | 0.202 | 0.707 | 17.293 | 0.854 | 0.142 | 0.088 | 0.907 | 18.293 | 0.894 | 0.112 |
| LGM | 0.144 | 0.800 | 17.643 | 0.869 | 0.158 | 0.138 | 0.823 | 17.893 | 0.884 | 0.139 |
| InstantMesh | 0.076 | 0.931 | 19.988 | 0.901 | 0.096 | 0.096 | 0.892 | 18.608 | 0.903 | 0.096 |
| PRM | 0.050 | 0.981 | 25.125 | 0.929 | 0.061 | 0.053 | 0.979 | 25.063 | 0.932 | 0.063 |

Table 2: Comparison with InstantMesh on GSO dataset: We used ground-truth rendered multi-view images as input. "Random m&r" indicates whether materials and lighting were randomly changed.

| Method | Random m&r | CD↓ | FS@0.1↑ | PSNR↑ | SSIM↑ | LPIPS↓ |
|---|---|---|---|---|---|---|
| InstantMesh | ✓ | 0.061 | 0.934 | 21.115 | 0.871 | 0.092 |
| InstantMesh | ✗ | 0.048 | 0.972 | 23.644 | 0.893 | 0.089 |
| PRM | ✓ | 0.053 | 0.982 | 24.602 | 0.921 | 0.065 |
| PRM | ✗ | 0.043 | 0.991 | 26.377 | 0.922 | 0.063 |

For both the specular lighting map $L_{\text{spec}}$ and the diffuse lighting map $L_{\text{diff}}$, which are directly influenced by surface normals, effectively optimizing these light maps significantly enhances the surface normals, thereby refining the precision of local details. This process is analogous to photometric stereo. The key difference is that these maps are exclusively related to surface normals without considering albedo, as demonstrated in Eq.(8) and Eq.(9), in contrast to shading color.

**Applications.** PRM achieves high-quality geometry reconstruction with predicted albedo. This capability enables us to render the object under novel lighting conditions and also to modify its material properties. Examples of these applications are provided in Figure 11 in the Appendix.

## 4 EXPERIMENTS

### 4.1 DATASETS AND EVALUATION PROTOCOL

**Datasets.** For training, we used Objaverse (Deitke et al., 2023), a dataset comprised of synthetic 3D assets, that allows us to control the lighting, geometry, and material properties. We first filter Objaverse to get a high-quality subset for training. The filtering process aims to exclude objects lacking texture maps or those of inferior quality, such as those with low-poly properties. Texture maps are essential as they provide detailed albedo maps; in their absence, vertex colors are used as a substitute for albedo. Moreover, low-poly meshes result in uneven surface lighting. During rendering, we maintained the original albedo unchanged and randomly selected a material combination from a total of 121 possibilities, which were derived by varying the metallic and roughness properties from 0 to 1 in increments of 0.1. For lighting, we utilized environment maps sampled from a collection of 679 maps available on Polyhaven.com, thereby ensuring a diverse range of lighting conditions.

For evaluation, We performed quantitative comparisons using two public datasets, including Google Scanned Objects (GSO) (Downs et al., 2022) and OmniObject3D (Omni3D) (Wu et al., 2023). We randomly picked out 300 objects as the evaluation set both datasets, respectively. To show the robust capabilities of our model on appearance variations, we rendered the input view of each object with a randomly sampled combination of materials and lighting. We also report extra comparison results, following previous methods that utilized fixed lighting and did not change materials.

**Evaluation Protocol.** We evaluated both the 2D visual quality and the 3D geometric quality. For the 2D visual evaluation, we rendered novel views from the reconstruced 3D mesh and compared them with the ground truth views, using PSNR, SSIM, and LPIPS as metrics. Since other methods do not predict albedo, we compared their shading color in novel views. For our method, we rendered the shading color based on the predicted mesh and albedo, then performed the metric calculations. For the 3D geometric evaluation, we first aligned the coordinate systems of the reconstruced meshes with those of the ground truth meshes. Subsequently, we repositioned and rescaled all meshes into a cube of size $[-1, 1]^3$. We reported the Chamfer Distance (CD) and F-Score (FS) at a threshold of 0.1, which were computed by uniformly sampling 16K points from the mesh surfaces.

### 4.2 IMPLEMENTATION DETAILS

Our model was developed based on InstantMesh (Xu et al., 2024). The architecture of the Transformer encoder, triplane transformer, and the FlexiCubes decoder mirrors that of InstantMesh. Our model underwent training for 7 days and 3 days on 32 NVIDIA RTX A800 GPUs for the first stage and second stage, respectively. For more details, please see our Appendix.

### 4.3 COMPARISON WITH STATE-OF-THE-ART METHODS

We compared the proposed PRM with four baselines. These include TripoSR (Tochilkin et al., 2024), an open-source LRM implementation renowned for its superior single-view reconstruction performance; CRM (Wang et al., 2024), a UNet-based Convolutional Reconstruction Model that reconstructs 3D meshes from generated multi-view images and canonical coordinate maps (CCMs); LGM (Tang et al., 2024), a unet-based Large Gaussian Model that reconstructs Gaussians from generated multi-view images; and InstantMesh (Xu et al., 2024), a transformer-based LRM that employs a two-stage training strategy for direct 3D mesh reconstruction. We reported both quantitative and qualitative comparative results for a complete comparison analysis.

**Quantitative Results.** We reported quantitative results with randomly selected lighting and materials in two different datasets in Table 1. We also report quantitative results without changing materials and using fixed lighting as previous methods compared with cutting-edge method Instantmesh on GSO in Table 2, where ground-truth rendered multi-view images were used as input.

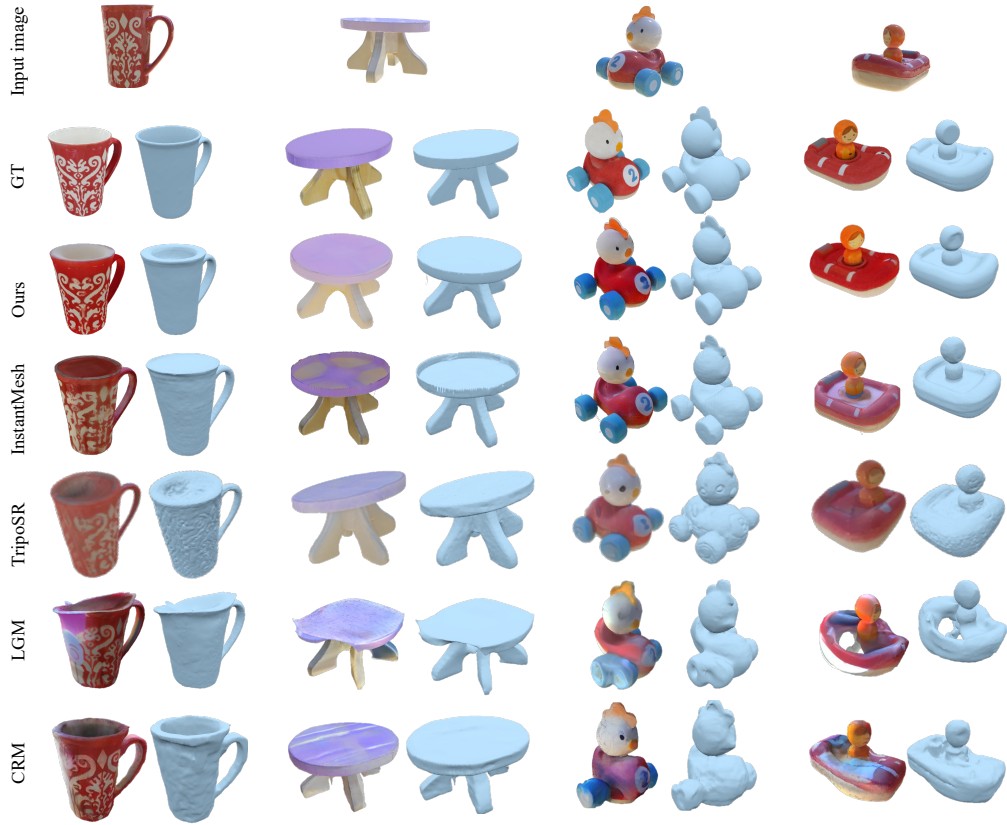

Figure 4: Qualitative comparisons with state-of-the art methods and ground truth for single-view reconstruction task. PRM reconstructs the highest quality 3D mesh and provides a more accurate texture prediction from input photographs compared to the others.

For 3D reconstruction metrics, PRM achieves significant improvements over all previous state-of-the-art methods on both datasets, as shown in Table 1. It records a relative 34% improvement, reducing the Chamfer Distance from 0.076 in InstantMesh to 0.050, and demonstrates substantial enhancement in FS@0.1, increasing from 0.931 in InstantMesh to 0.981 on the GSO dataset. The qualitative comparison with other methods is presented in Figure 4. We attribute these improvements to our use of photometric stereo images for both input and supervision. This approach not only enables the model to learn fine-grained geometric details by providing rich photometric cues but also enhances the model's robustness to variations in image appearance. Further validation of these results in the test setting without material changes and using fixed lighting is shown in Table 2.

For 2D visual metrics, our approach effectively mitigates the impact of lighting variations to accurately restore the original colors of objects, as shown in Table 1. We surpass all current methods across all metrics. For example, on the GSO dataset, our PSNR has improved from 19.988 to 25.125, SSIM from 0.901 to 0.929, and LPIPS has decreased from 0.096 to 0.061. Similar performance gains are also observed on the OmniObject3D dataset.

**Qualitative Results.** For the qualitative comparison, we randomly selected four images from the GSO dataset to serve as inputs for 3D model reconstruction. For each reconstructed mesh, we visualized both the albedo (ours) and the shading color (others), as well as the pure geometry.

As shown in Figure 4, the results reconstructed by PRM exhibit significantly more accurate geometry and appearance. Our model can reconstruct precise geometry and accurately predict albedo from images with specular highlights, whereas other methods fail. For instance, InstantMesh often predicts uneven geometric surfaces and tends to reconstruct incorrect geometry. TripoSR, on the other hand, frequently confuses texture and lighting information with geometric details, leading to erroneous final reconstructions. Similarly, both CRM and LGM struggle to produce satisfactory results, falling short in both geometry accuracy and texture prediction. This underscores the robustness and superior performance of our PRM method in handling complex lighting conditions and intricate surface details, making it a more reliable choice for high-quality 3D model reconstruction.

## 4.4 ABLATION STUDY

We conducted the ablation study to validate the effectiveness of each component in our framework.

**The effectiveness of PBR.** PBR plays a crucial role in improving geometry, especially for glossy surfaces. To validate the effectiveness of PBR, we conducted experiments by directly predicting shading colors using an MLP, rather than deriving results with PBR. It is important to note that we continued to utilize our rendering method with varied materials and lighting, which demonstrates significant view-dependent effects. However, direct prediction of shading color, without accounting for view-dependent appearances, struggles to model such effects and thus fails to accurately reconstruct geometry. We presented these results as "w/o PBR" in Figure 8.

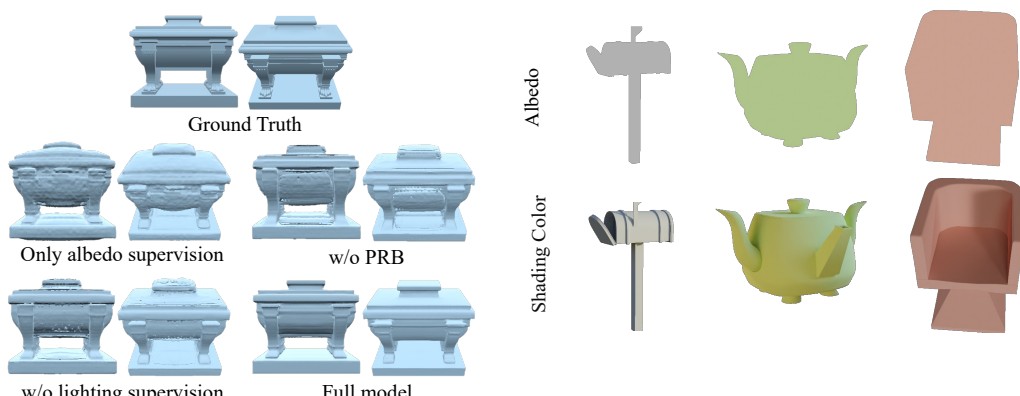

Figure 5: Ablation study to validate the effectiveness of each individual component.

Figure 6: Shading color offers significant photometric cues for perceiving geometry, whereas albedo lacks this information.

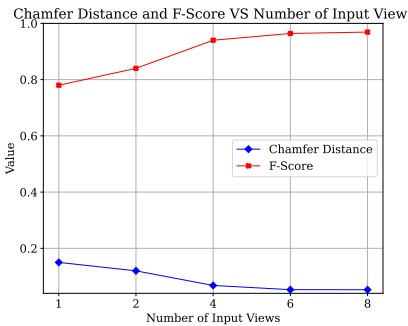

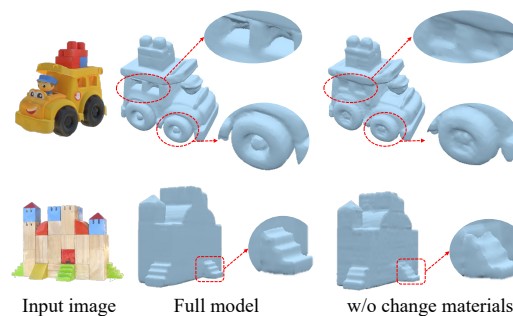

Input image    Full model    w/o change materials

Figure 7: Ablation study of the effect of number of input views.

Figure 8: Ablation study of the effect of changing materials during training.

**Albedo supervision only.** To avoid the interference caused by specular color on the surface, an intuitive approach is to directly use albedo instead of shading color for supervision. However, this method proves ineffective in practice, as albedo contains few photometric cues, thereby hindering geometry reconstruction. For example, a concave surface with uniform albedo may appear planar without the presence of cast shadows, while shading color provides significant photometric cues critical for accurately perceiving geometry, as illustrated in Figure 6. We conducted an ablation study within our framework by excluding shading color and light maps from supervision. The qualitative comparison is shown in Figure 5 "Only albedo supervision". The results demonstrate that photometric cues are crucial for accurate geometry reconstruction.

**Lighting maps supervision.** We conducted the ablation study within our framework by excluding the lighting maps loss. The results, denoted as "w/o lighting supervision", are displayed in Figure 5 . Since lighting maps are exclusively related to surface normals and do not include albedo, optimizing these maps proves particularly beneficial for enhancing the optimization of fine-grained local details.

**The number of camera views.** We conducted an ablation study to illustrate the importance of varying camera poses as input during training. The quantitative results, including Chamfer Distance (CD) and F-Score, are depicted in Figure 7 by varying number of camera views from 1 to 8. When more images rendered under different camera views are inputted, we achieve better results. Additional visualization results can be found in Figure 14 in the Appendix .

**Variations in materials.** We conducted an ablation study to illustrate the effectiveness of varying materials during training. The qualitative results are shown in Figure 8. Without changing the materials, some details are lost, as varying materials leads to a greater number of equations for an optimal solution. Moreover, the model has lost the capability to reconstruct glossy surfaces.

## 5    CONCLUSION AND LIMITATION

**Limitation.** Despite the high-quality results achieved in this work, there remain several limitations for future research to explore: 1) Firstly, the reconstructed 3D model is sensitive to the quality of multi-view images. If the pre-trained multi-view diffusion model performs poorly in converting single views to multi-view images, our model may produce sub-optimal results as shown in Figure 15 in the Appendix. 2) Secondly, the accuracy of the estimated albedo appears to be somewhat entangled with the lighting conditions.

**Conclusion.** In this work, we introduce PRM, a novel feed-forward framework designed to reconstructed high-quality 3D assets that feature fine-grained local details, even amidst complex image appearances. To achieve this goal, we utilize photometric stereo images as both input and supervision, providing sufficient photometric cues for fine-grained local geometry recovery and enhancing the model's robustness to variations in image appearance. Using a mesh as our 3D representation, we employ differentiable PBR for predictive rendering, underpinning the utilization of multiple photometric supervisions for optimization. Experiments on public datasets validate that PRM surpasses other methods by a large margin.

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

# A APPENDIX

## A.1 BRDF PARAMETERIZATION

In Sec. 3.1 we introduce the $D$, $F$ and $G$ term of the specular component of BRDF property. We implement the Cook-Torrance BRDF model (Cook & Torrance, 1982). The basic specular albedo $F_0 = (m * \boldsymbol{a} + (1 - m) * 0.04)$, where $\boldsymbol{a}$ is the albedo and $m$ is the metalness. The Fresnel term ($F$) is defined as:

$$F = F_0 + (1 - F_0)(1 - (\boldsymbol{h} \cdot \boldsymbol{\omega}_o))^5, \tag{13}$$

where $\boldsymbol{h}$ is the half-way vector between $\omega_o$ and viewing direction $\boldsymbol{\omega}_i$. The normal distribution function $D$ is Trowbridge-Reitz GGX distribution as

$$D(h) = \frac{\alpha^2}{\pi \left( (\mathbf{n} \cdot \mathbf{h})^2 (\alpha^2 - 1) + 1 \right)^2}, \tag{14}$$

where $\alpha = \rho^2$, $\mathbf{n}$ is the surface normal. The geometry term $G$ is the Schlick-GGX Geometry function:

$$G(\boldsymbol{n}, \boldsymbol{\omega}_o, \boldsymbol{\omega}_i, k) = G_{\text{sub}}(\boldsymbol{n}, \boldsymbol{\omega}_o, k) G_{\text{sub}}(\boldsymbol{n}, \boldsymbol{\omega}_i, k), \tag{15}$$

where $G_{\text{sub}}$ is given by:

$$G_{\text{sub}}(\boldsymbol{n}, \boldsymbol{\omega}, k) = \frac{\boldsymbol{n} \cdot \boldsymbol{\omega}}{(\boldsymbol{n} \cdot \boldsymbol{\omega})(1 - k) + k}, \tag{16}$$

where $k$ is a parameter related to the roughness $\rho$, often approximated as $k = \frac{\rho^4}{2}$.

## A.2 OPTIMIZATION AND ADDITIONAL MODEL DETAILS

**Optimization Details.** We used Adam (Kingma & Ba, 2014) as our optimizer. In the first stage, the learning rate was set to $4 \times 10^{-5}$. In the second stage, the learning rate was set to $4 \times 10^{-6}$ for finetuning. We used 32 NVIDIA A800 GPUs in the first stage for nerf training with a batch size of 256 for 100K steps, taking about 7 days. In the second stage, We used 32 NVIDIA A800 GPUs to finetune the model from the first stage with a batch size of 256 for 30K steps, taking about 3 days.

**Network architecture.** Our network architecture is similar to that of InstantMesh (Xu et al., 2024), consisting of a pre-trained DINO that encodes images into image tokens, and an image-to-triplane transformer decoder that projects these 2D image tokens onto a 3D triplane using cross-attention. Furthermore, three MLPs are utilized, taking interpolated triplane features as input and outputting albedo, SDF, deformation, and weights. These outputs are required by FlexiCube for mesh extraction and subsequent rendering. The details of the network is shown in Figure 9 Our final model is a large transformer with 16 attention layers, with feature dimension 1024. The size of triplane is $64 \times 64 \times 3$ with 80 channels. The grid size for FlexiCube was set to 128. The resolution of input images was 512.

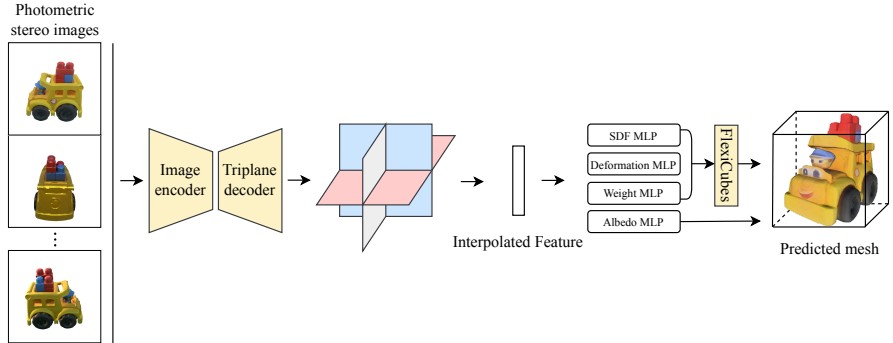

Figure 9: The details of network architecture.

## A.3 QUANTITATIVE RESULTS OF ABLATION STUDY

We reported quantitative results of our ablation study in Table 3. Due to the high computation cost for each trial, it is infeasible for us to include all training objects for ablation study. Alternatively, we used 10k training objects for our ablation study. The evaluation set is the same as we did in the main experiments.

Table 3: Quantitative results of our ablation study.

| Metric | CD↓ | FS@0.1↑ | PSNR↑ | SSIM↑ | LPIPS↓ |
|---|---|---|---|---|---|
| Only albedo supervision | 0.099 | 0.887 | 17.532 | 0.795 | 0.188 |
| w/o PBR | 0.089 | 0.909 | 19.143 | 0.724 | 0.154 |
| w/o lighting supervision | 0.073 | 0.919 | 20.114 | 0.810 | 0.155 |
| w/o change materials | 0.089 | 0.894 | 19.662 | 0.817 | 0.159 |
| Full model | 0.066 | 0.948 | 20.992 | 0.830 | 0.137 |

## A.4 TRAINING STRATEGY

**Camera Augmentation.** Previous LRMs typically prepare training data by rendering images with fixed Fields-Of-View (FOVs) and camera distances, making the models sensitive to changes in these variables during inference. Since we adopt a real-time rendering method and mesh rasterization for fast online rendering, we can readily adjust the FOVs and camera distances during training. This training strategy enhances our model's robustness to variations in camera embeddings. We provide some examples in the following section.

**Random materials and lighting.** During inference, one option for 3D mesh reconstruction is to leverage a multi-view diffusion model to generate multi-view images. However, these images may exhibit inconsistencies in materials or lighting. To ensure our model remains robust to these inconsistencies, we randomly change the materials and lighting when rendering each view during training. Alternatively, the lighting and materials of the input images are consistent. Our model need to handle this scenario. Therefore, we establish a threshold to ensure that the rendered multi-view images potentially share the same materials and lighting. Specifically, when rendering each view, there is a 50% probability that the materials and lighting will change. This arrangement means that each view may feature different materials or lighting. If no changes are made, all views are rendered with consistent materials and lighting.

## A.5 EXAMPLE OF PHOTOMETRIC STEREO IMAGES

In this section, we present examples of rendered photometric stereo images along with intermediate shading variables, including specular lighting, diffuse lighting, albedo maps and environment maps, as shown in Figure 10. The red box highlights how varying roughness levels influence the specular lighting maps, affecting their frequency. Specifically, lower roughness (right) results in specular lighting of higher frequency.

## A.6 APPLICATION VISUALIZATION

Since our method can reconstruct high-quality meshes with predicted albedo, it facilitates downstream applications such as relighting and material editing. We showcase some examples in Figure 11.

## A.7 ROBUSTNESS EVALUATION

**Robustness to Camera Embedding.** PRM exhibits robustness to variations in camera embedding. We compared PRM with InstantMesh by altering the camera embedding (i.e., FOV and camera radius) during inference. The results, shown in Figure 12, demonstrate that PRM maintains strong robustness to changes in camera embedding, whereas the performance of InstantMesh declines significantly when camera embedding varies.

**Robustness to image appearance.** PRM is robust to the image appearance. When handling specular surfaces, we can achieve correct geometry reconstruction. More visualization results can be found in Figure 13.

**Robustness to spatially-varying materials.** PRM is robust to the objects with spatially-varying materials for both synthetic and real-captured images. More visualization results can be found in Figure 18.

## A.8 THE EFFECT OF THE NUMBER OF CAMERA VIEWS

We demonstrate the importance of varying camera poses for rendering multi-view photometric stereo images as input. The number of input views is increased from 1 to 8. The qualitative results are illustrated in Figure 14. Better results are achieved with more input views; using 4 or 6 views provides the optimal balance between effectiveness and efficiency.

## A.9 FAILURE CASES

Although effective, the performance of PRM is constrained by the quality of the multi-view images generated by the multi-view diffusion model when performing single image to 3D tasks. We illustrate a failure case in Figure 15. The lack of depth information in the input image leads to undesirable multi-view image generation, resulting in a reconstructed 3D mesh that lacks accuracy. A potential solution is to use the estimated depth to guide the multi-view images generation. We show an example of the estimated depth by DepthAnythingV2 (Yang et al., 2024) in Figure 16. Our method cannot handle multi-view images with background as shown in Figure 19, since we used images with while background as input during training as previous methods do. However, we can easily obtain images with while background by pretrained segmentation model.

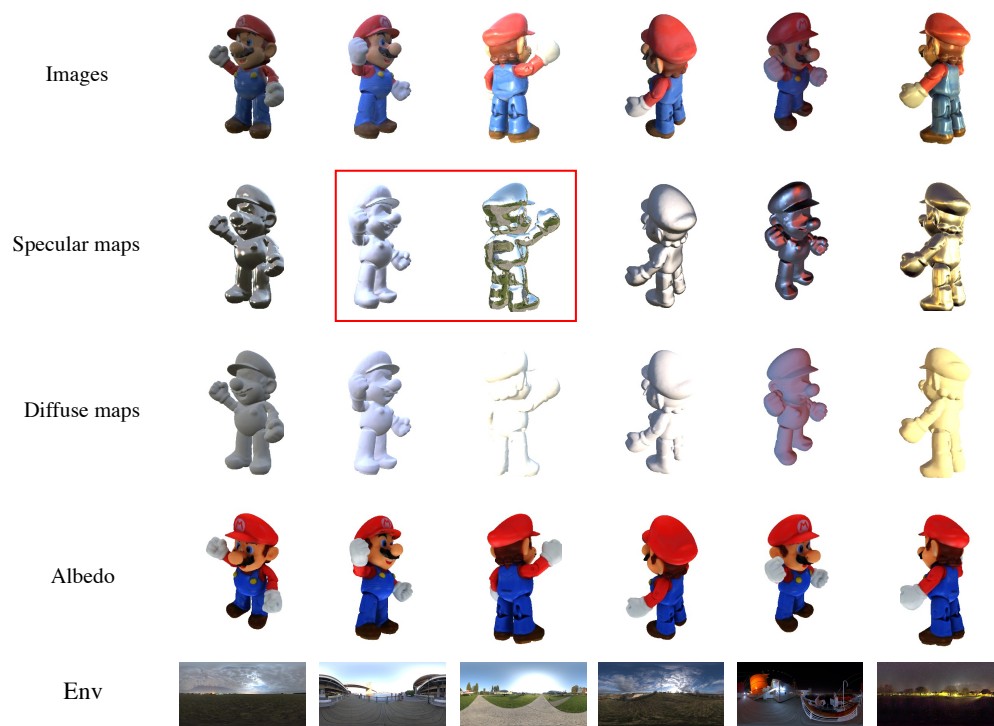

Figure 10: Examples of rendered photometric stereo images, along with specular, diffuse lighting maps and albedo maps. The red box highlights how varying roughness levels influence the specular lighting maps, affecting their frequency. Specifically, lower roughness (right) results in specular lighting of higher frequency.

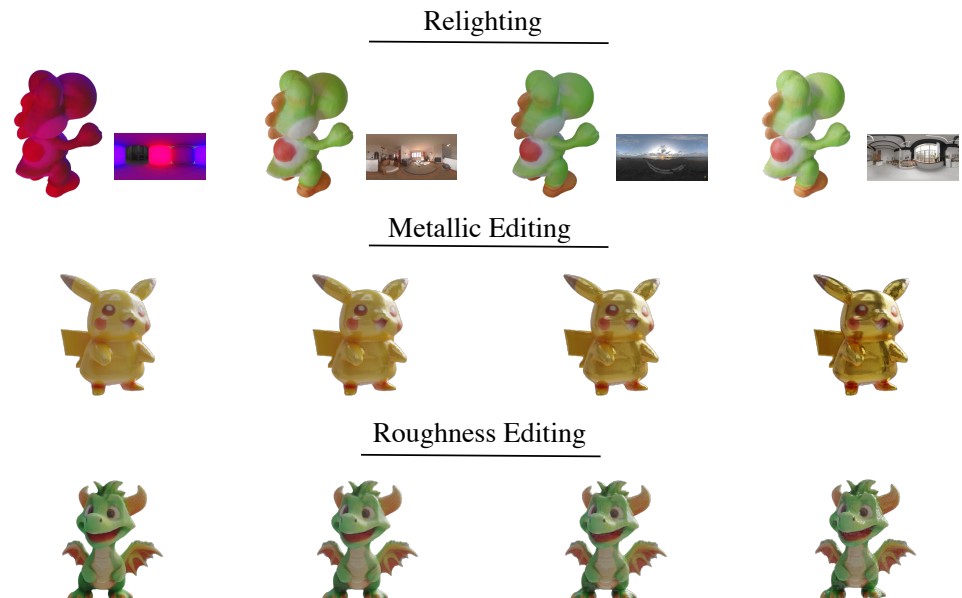

Figure 11: Application visualization. We show relighting and materials editing here.

## A.10   MORE VISUALIZATION RESULTS

We show more visualization results of PRM in Figure 17.

## A.11   QUALITATIVE COMPARISON WITH MESHLRM

We also show some qualitative comparisons with MeshLRM (Wei et al., 2024) in Figure 20. PRM achieves better reconstruction performance on both glossy surfaces and complex objects.

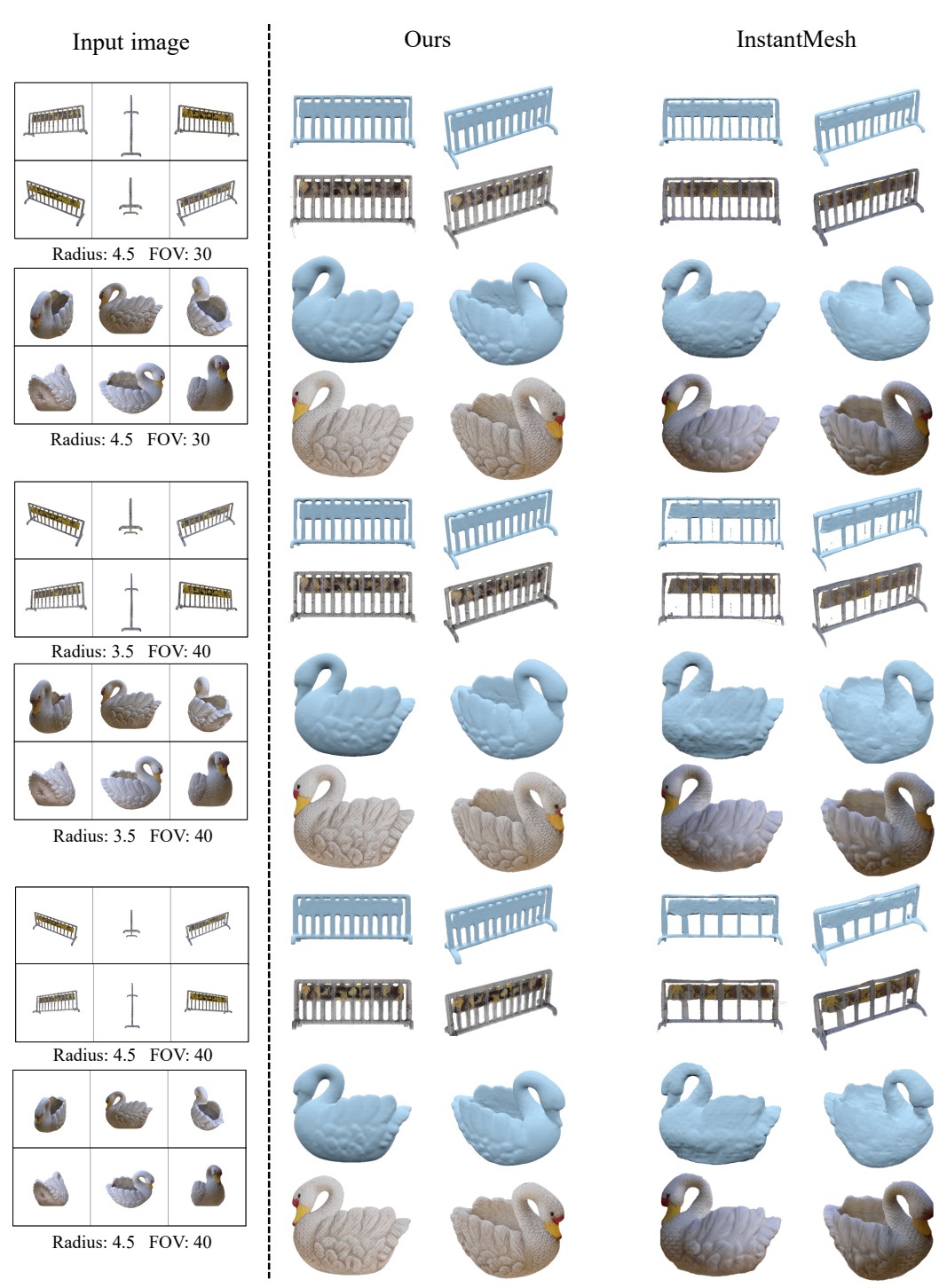

Figure 12: Comparison with InstantMesh when changing FOVs and camera radius: PRM demonstrates robustness to variations in camera embedding. Conversely, InstantMesh struggles when the radius and FOV differ from those used during training.

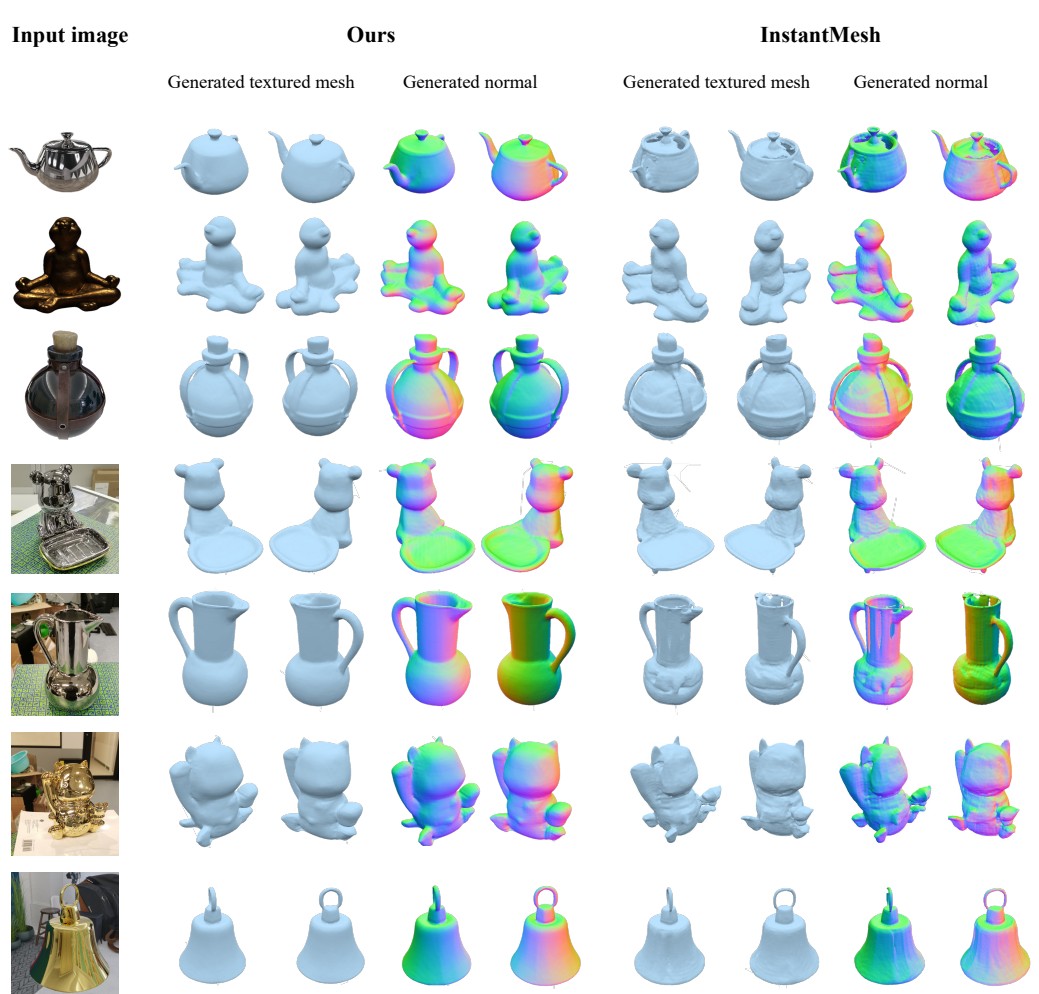

Figure 13: Single view reconstruction results using our method on input images with extreme conditions, such as specular highlights and shadows. Despite challenging lighting conditions, PRM successfully reconstructs the geometry and surface normals with high fidelity.

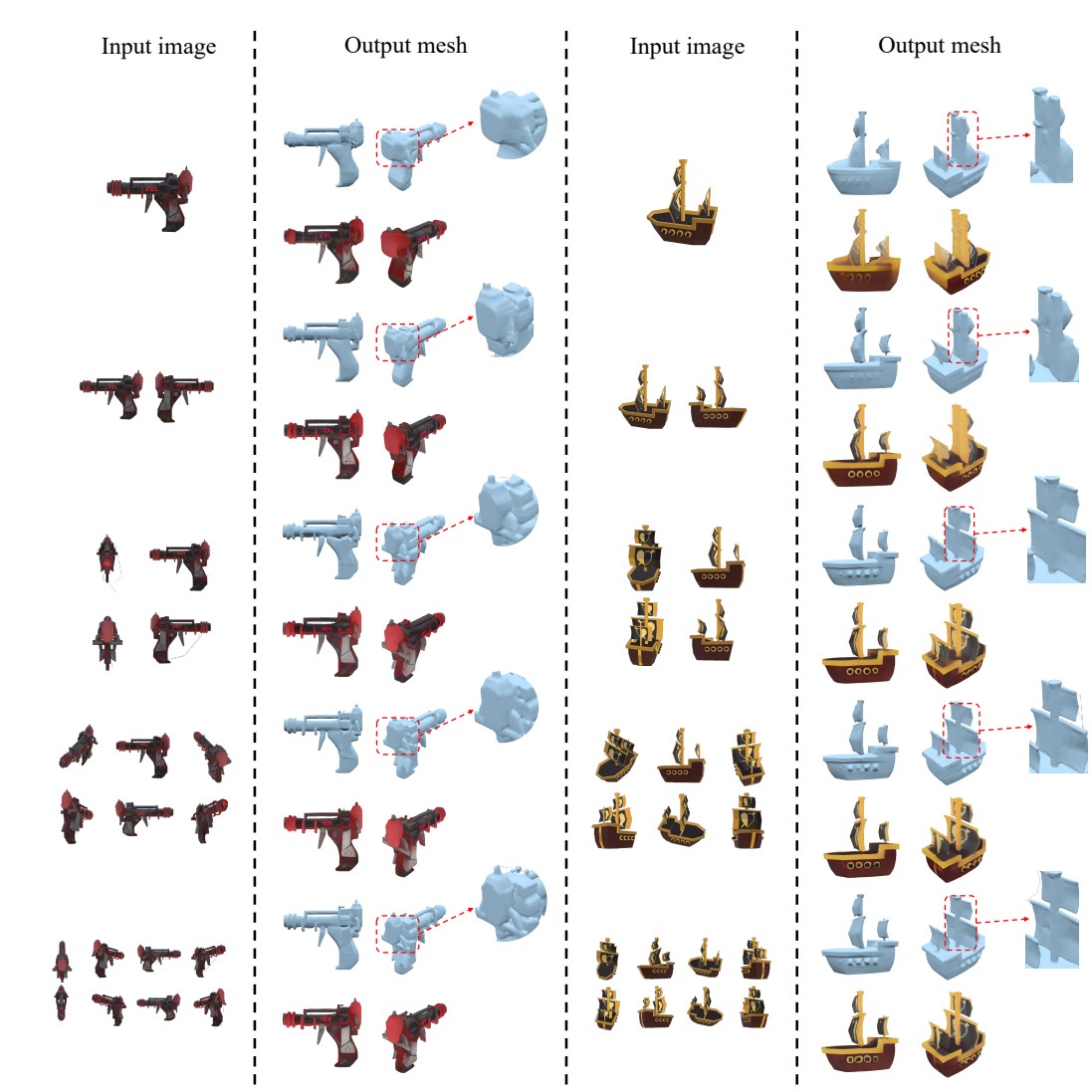

Figure 14: The effect of the number of input views. More views lead to better reconstruction result.

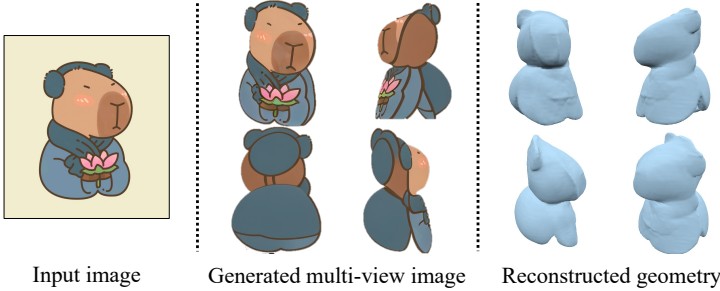

Figure 15: Illustration of a failure case.

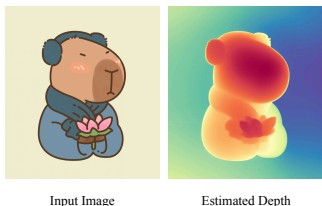

Input Image          Estimated Depth

Figure 16: DepthAnythingV2 can estimate correct depth for image that lacks depth information, which may help multi-view diffusion model generate more reasonable multi-view images.

| Input image | Generated mesh | Input image | Generated mesh |
|---|---|---|---|

Figure 17: Visualization of more results of single view to 3D task.

Input image   Material map   Generated textured mesh        Generated normal              Generated albedo

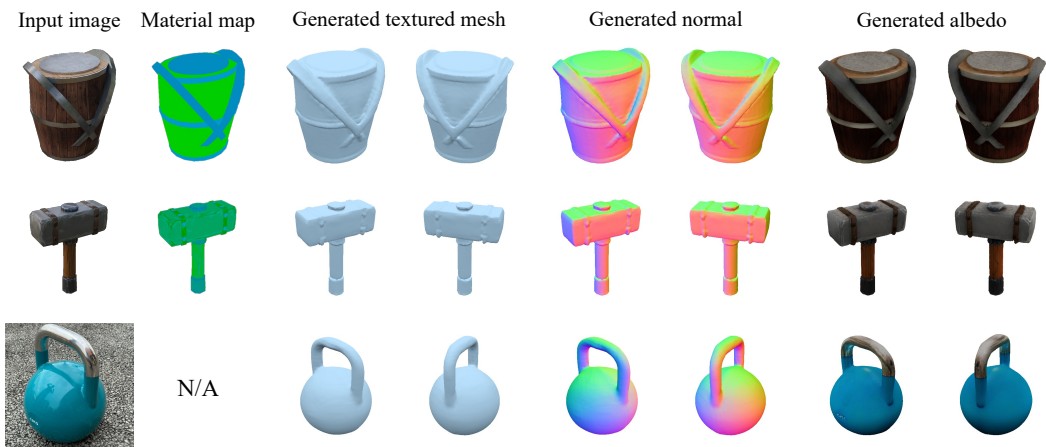

Figure 18: PRM can handle objects with spatially-varying materials for both synthetic and real-captured images.

Input image

Generated color mesh

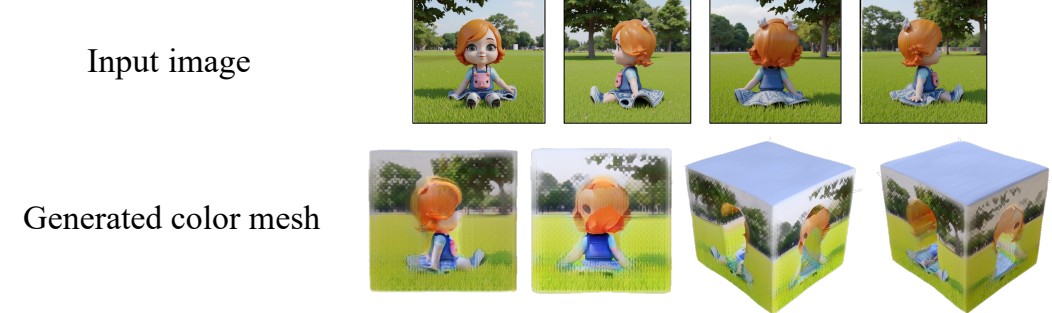

Figure 19: Our method fails to handle images with natural background since we takes images with while background as input during training.

Input image                    Ours                        MeshLRM

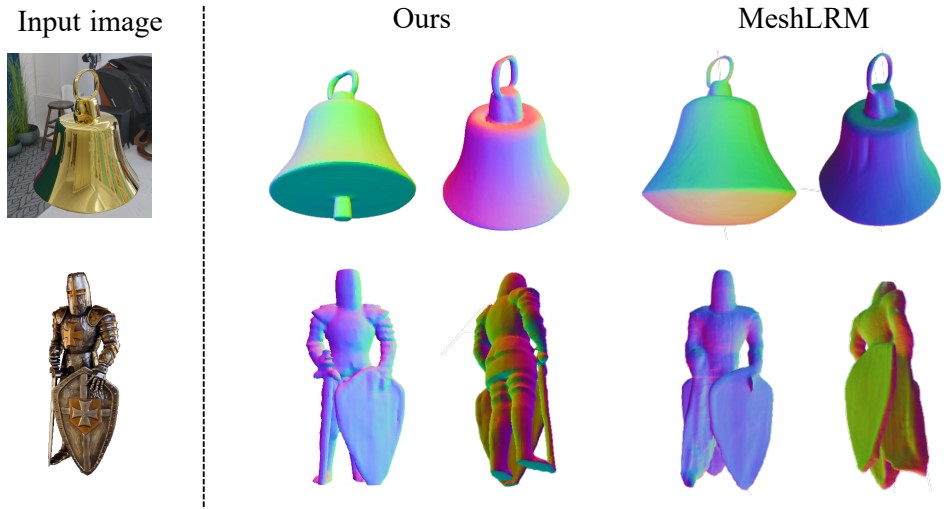

Figure 20: Qualitative comparison with MeshLRM.

