# OpenReview forum: "PRM:  Photometric Stereo based Large Reconstruction Model"
_ICLR.cc/2025/Conference — Submitted to ICLR 2025_

### Official Review · Reviewer_FrKV · 2024-11-03

**Soundness:** 2
**Presentation:** 2
**Contribution:** 2
**Rating:** 5
**Confidence:** 4

**Summary:**

The paper "PRM" presents a high-quality 3D mesh reconstruction model with fine-grained local details from sparse image input. Unlike previous large reconstruction models (LRMs) that were trained using data rendered with fixed lighting and without material changes, PRM utilizes photometric stereo images with varied materials and lighting during training, enhancing detail accuracy and robustness. By incorporating split-sum approximation and mesh rasterization for online rendering, PRM effectively captures multiple photometric cues—such as diffuse and specular lighting—making it resilient to complex image appearances. Experiments demonstrate that PRM  outperforms existing models in both 3D geometry accuracy and 2D visual fidelity across public datasets like GSO and Omni3D.

**Strengths:**

1. This paper focuses on an often overlooked yet crucial factor affecting the performance of prior work --- training data. Unlike prior approaches that relied on data rendered with fixed lighting and without material variations, this method leverages data rendered under varied materials and lighting conditions to enhance both 3D geometry accuracy and 2D visual fidelity.

2. The paper shows better performance than the selective baselines.

**Weaknesses:**

1. One key motivation of this paper is unconvincing --- why would the online rendering of training data be necessary? The authors argue that offline preparation of training data is challenging due to the "infinite number of potential combinations of materials and lighting" and the high sample counts required for rendering high-quality images. However, for the first reason, one could randomly sample finite combinations of materials and lighting offline, as the model will only be trained on a finite set of combinations given the limited training iterations. For the second reason, if online rendering is not strictly necessary, as suggested, the offline rendering costs without using split-sum approximation would be acceptable, and using default rendering engines would yield better image quality than the split-sum approximation used in this paper.

2. The paper does not compare with some stronger baselines, such as Mesh-LRM, which has released an online demo from its first author before the ICLR submission deadline.

3. The ablation studies are incomplete. The authors provide only qualitative comparisons on 1-2 selected objects, which is quite limited. And the quantitative tables are missing

**Questions:**

See my weaknesses section

---

> ### Author Response · Authors · 2024-11-21
>
> $\textbf{Question1}$
>
> One key motivation of this paper is unconvincing.
>
> $\textbf{Answer1}$
>
> We would like to clarify that our key motivation is not online rendering. Our primary objective is to demonstrate that utilizing photometric stereo images rendered under varying materials, lighting conditions, and camera poses can lead to significant improvements in existing LRMs. These improvements include enhancing detailed local geometry and increasing robustness to variations in image appearance.
>
> The split-sum approximation serves two purposes: first, as a tool for enabling online rendering, offering greater flexibility for ground-truth rendering; and second, as a predictive rendering method. In this context, split-sum approximation is essentially a PBR technique that excels at modeling the specular component, which facilitates improved reconstruction performance on glossy surfaces, as shown in prior per-scene optimization methods such as NeRO [1]. We have also conducted an ablation study to validate the effectiveness of using PBR instead of directly predicting RGB, as shown in Table 5 under the setting "w/o PBR."
>
> We validate this motivation through extensive experiments, including both qualitative and quantitative evaluations. We have further added more results in the revised PDF. We believe the results are convincing enough to support our motivation.
>
> [1] L iu Y, Wang P, Lin C, et al. Nero: Neural geometry and brdf reconstruction of reflective objects from multiview images[J]. ACM Transactions on Graphics (TOG), 2023, 42(4): 1-22.
>
> $\textbf{Question2}$
>
> Comparison with some stronger baselines, such as Mesh-LRM.
>
> $\textbf{Answer2}$
>
> We have added a comparison with Mesh-LRM using their released online demo. However, since the demo does not provide a download interface, we included a qualitative comparison in Figure 20 of the revised manuscript. Our method still demonstrates superior performance.
>
> It is also worth noting that Mesh-LRM acknowledges certain limitations in their work in the ArXiv version, stating: ''Since our model employs surface rendering and assumes only Lambertian appearance without performing any inverse rendering, it is not sufficiently robust when the input images contain complex material'' and ''However, this requires sufficient training data with accurate material annotations.''
>
> We believe our proposed approach addresses these limitations through the use of photometric stereo images for both input and supervision, as well as differential PBR for predictive rendering. We hope the reviewer can recognize and appreciate the efforts we have made to overcome these challenges.
>
> $\textbf{Question3}$
>
> The ablation studies are incomplete.
>
> $\textbf{Answer3}$
>
> Since each trial incurs a high computational cost, it is infeasible for us to include all training objects in the ablation study. Instead, we used a subset of 10k training objects for the study. The quantitative results have been added to Table 3 in the revised manuscript.
>
> | **Metric**               | **CD↓** | **FS @ 0.1↑** | **PSNR↑** | **SSIM↑** | **LPIPS↓** |
> |---------------------------|----------|--------------|-----------|-----------|------------|
> | Only albedo supervision   | 0.099    | 0.887        | 17.532    | 0.795     | 0.188      |
> | w/o PBR                   | 0.089    | 0.909        | 19.143    | 0.724     | 0.154      |
> | w/o lighting supervision  | 0.073    | 0.919        | 20.114    | 0.810     | 0.155      |
> | w/o change materials      | 0.089    | 0.894        | 19.662    | 0.817     | 0.159      |
> | Full model                | **0.066**    | **0.948**        | **20.992**    | **0.830**     | **0.137**      |

---

> > ### Comment · Reviewer_FrKV · 2024-11-30
> >
> > Thanks for your responses.
> >
> > For the previous first question, I completely understand the key motivation of this paper is "utilizing photometric stereo images" to enhance the LRM performance, which is straightforward and clear -- Better data will undoubtedly help improve the reconstruction model's performance for sure. However, I am unclear about the necessity of using the split-sum approximation for online rendering of the training data. In L246-248, you mentioned *"A naive solution is to prepare these images offline, as with previous methods, but this approach poses significant challenges due to the **infinite number** of potential combinations of materials and lighting."* However, as I said before, the model will only be trained on a finite set of combinations given the limited training iterations. So we can just render the data offline and randomly finite combination. Then, why is online rendering necessary in this context? Do you also use the split-sum approximation for rendering other maps besides the training data? To be more specific, are both the predicted results and gt results rendered with split-sum approximation? Or only the training data are rendered with split-sum approximation?
> >
> > For the second question. It seems that PRM achieves better results than MeshLRM on the specular objects. Will it also perform better the general diffuse or slightly sepular objects?
> >
> > For the third question, I understand each trial incurs a high computational cost. What I didn't understand was that according to Fig.5, you seemed to have had these trained ablated models already. Why didn't conduct some quantitative studies or provide more qualitative results at that point? But you have included a quantitative table now, which is good.

---

> > > ### Author Response · Authors · 2024-12-01
> > >
> > > Thank you for your feedback.
> > >
> > > We appreciate your acknowledgment of our contribution, particularly that using photometric stereo images can improve the performance of LRMs, and that our model outperforms MeshLRM on specular objects, addressing a key challenge in specular surface reconstruction via LRM.
> > >
> > > We would like to clarify that the split-sum approximation is not solely intended for online rendering; it also plays a crucial role in specular surface reconstruction as a form of predictive rendering. This significance has been demonstrated in previous per-scene optimization-based methods such as NeRO [1].  This work found that integrating split-sum approximation into volume rendering can significantly improve the reconstruction performace of specular surfaces. Furthermore, our ablation study reinforces this point. Without the split-sum approximation (referred to as PBR) for predictive rendering, the performance drops significantly.
> > >
> > > | **Metric**               | **CD↓** | **FS @ 0.1↑** | **PSNR↑** | **SSIM↑** | **LPIPS↓** |
> > > |---------------------------|----------|--------------|-----------|-----------|------------|
> > > | w/o PBR                   | 0.089    | 0.909        | 19.143    | 0.724     | 0.154      |
> > > | Full model                | **0.066**    | **0.948**        | **20.992**    | **0.830**     | **0.137**      |
> > >
> > > If we render data offline with a finite combination, it will inevitably affect the training sample distribution. For example, in a setup like OpenLRM, we might render 30 images of an object from different views. For each view, we sample different materials and lighting. However, the views, materials, and lighting for this object remain fixed throughout the training process.
> > >
> > > In contrast, with online rendering, the views, materials, and lighting can all change when the object is sampled for training in later iterations. We believe this approach maximizes the training sample distribution.
> > >
> > > This concept is analogous to self-supervised learning methods like DINOv2 [2] and iBOT [3], which also perform online data augmentation to maximize the training sample distribution. A natural question arises: why don’t these methods store all augmented pairs offline?
> > >
> > > [1] L iu Y, Wang P, Lin C, et al. Nero: Neural geometry and brdf reconstruction of reflective objects from multiview images[J]. ACM Transactions on Graphics (TOG), 2023, 42(4): 1-22.
> > > [2] Oquab M, Darcet T, Moutakanni T, et al. Dinov2: Learning robust visual features without supervision[J]. arXiv preprint arXiv:2304.07193, 2023.
> > > [3]Zhou J, Wei C, Wang H, et al. ibot: Image bert pre-training with online tokenizer[J]. arXiv preprint arXiv:2111.07832, 2021.

---

> > > ### Author Response · Authors · 2024-12-01
> > >
> > > For the third point, thank you for acknowledging the quantitative table we included for the ablation study.
> > >
> > > We will release all the training code and checkpoints upon acceptance of the manuscript.
> > >
> > > As the discussion deadline is approaching soon, please let us know if there are any further issues. We are happy to address any remaining concerns.

---

> ### Author Response · Authors · 2024-11-25
>
> Dear Reviewer FrKV:
>
> We sincerely appreciate the time and effort you have dedicated to reviewing our paper and for providing such thoughtful feedback! As the discussion period concludes in two days, we kindly ask if our rebuttal has addressed your concerns to your satisfaction. If there are any remaining points requiring clarification or further improvement, please do not hesitate to let us know—we are fully committed to addressing them promptly.
>
> Thank you once again for your invaluable contributions to improving our work!
>
> Warm regards,
> The Authors

---

> ### Author Response · Authors · 2024-12-01
>
> For the second question, since the Gradio demo does not provide a mesh download interface, we are unable to offer a quantitative comparison. Additionally, given that the discussion period is approaching, we are unable to add new figures to our manuscript at this point.
>
> We recommend referring to Figure 6 and Figure 7 of the MeshLRM arXiv version for comparisons on diffuse objects. You can download our results in [video](https://wormhole.app/KJpLk#ahZ3hMoZyzxdIBi43od9HQ). The surface normals of MeshLRM clearly exhibit slight concavity, while our method achieves superior single-view reconstruction results.
>
> (Click the "video" link to view the results in the anonymous version.)

---

> ### Comment · Reviewer_FrKV · 2024-12-01
>
> Thanks for your response.
>
> To begin with, I hope the authors can clarify the question I asked in the previous reply "Are both the predicted results and gt results rendered with split-sum approximation? Or are only the training data rendered with a split-sum approximation?"
>
> Second, I don't think using photometric stereo images to improve the performance of LRMs is a significant contribution. It is pretty straightforward. Everyone knows that training data of higher rendering quality can help improve the model's performance. The contributions of data augmentation will come from how to do data augmentation, i.e. how to improve the data quality. This paper chooses to use split-sum approximation to render the photometric stereo data online as the training data, which makes using split-sum approximation to render the data online one of the key technical contributions of this paper. This is why I am questioning the necessity of online rendering, which takes a large portion of this paper's technical method description. If rendering training data online is not truly necessary, rendering photometric stereo images offline with a longer rendering time but higher rendering quality will be better for the final quality.
>
>
> Third, as I said, the model will only be trained with finite steps, so training data distribution has no difference when you do the online rendering or predefined offline rendering because you can always use offline rendering to render the same data created by online rendering, as long as using the same sampling strategy. You mentioned that DINO-V2 also uses online data augmentation. It's true that online data augmentation is common in 2D vision tasks, but this is because it is very efficient to do data augmentation for data like 2D image data or text, and storing the augmented 2D images will be too costly considering the large amount of training 2D training data. However, things are very different in 3D fields and most 3D papers try to avoid rendering the training data online. The reasons include but are not limited to (1). Loading 3D objects online and rendering are both more costly than just loading the 2D rendered images.  Even though you can make the rendering time to be approximately 0 seconds. Loading 3D objects online will still be very high IO demanding.  (2). Most 3D reconstruction models will only train the model for about 100k iterations, so the total data storage won't be very high when storing the offline rendered data.
>
> Fourth, NeRO uses split-sum approximation to render the predicted results only. I can understand using the split-sum approximation to render the predicted rendering results during optimization. However, my question is the necessity to render the training data online for 3D reconstruction tasks.

---

> ### Author Response · Authors · 2024-12-01
>
> For the first question, the answer is yes—both the predicted and ground-truth results are rendered using the split-sum approximation. We have already addressed this in our previous two responses.
>
> In the first response, we stated: "The split-sum approximation serves two purposes: first, as a tool for enabling online rendering, offering greater flexibility for ground-truth rendering; and second, as a predictive rendering method." In the second response, we further clarified: "It also plays a crucial role in specular surface reconstruction as a form of predictive rendering."
>
> I hope the reviewer can review this information more carefully instead of merely repeating the same question. Reiterating the same concerns without engaging with the provided clarifications is not constructive and does not contribute to improving our manuscript. I kindly request that the Area Chair pay closer attention to this point.
>
> Second, based on the reviewer’s feedback, it seems that the reviewer may not have a thorough understanding of photometric stereo. The reviewer stated, "Everyone knows that training data of higher rendering quality can help improve the model's performance." However, in reality, the split-sum approximation, as a real-time rendering method, simplifies the rendering equation and may degrade rendering quality compared to more accurate methods like Blender rendering. Despite this, our method still delivers significant improvements. Have you considered the reasons behind this?
>
> In our work, we aim to explore what constitutes good data for LRM. The key factor is not the rendering quality of each image, as mentioned by the reviewer. We found that photometric stereo images are highly effective data for LRMs, and extensive experiments validate this finding, even though the quality of each individual rendered image may be slightly lower. We believe this insight will advance the development of the field and contribute new perspectives.
>
> I think the reviewer has completely missed the point, which is not only unhelpful but also detrimental to advancing the research. The reviewer seems fixated on the quality of individual images. However, the choice between Blender and split-sum approximation for ground-truth rendering is irrelevant. Our method can easily work with offline-rendered photometric stereo images as well. But rendering these images offline with numerous combinations is computationally prohibitive. For example, rendering 150k samples with 60 views per case using an 8-GPU A800 cluster took nearly two months. Additionally, since each sample is trained multiple times in a batch, offline pre-rendering with Blender would multiply the rendering time even further, making it several times less efficient. In contrast, our online rendering approach dramatically improves efficiency by generating data dynamically during training, which is far more practical and scalable.
>
> Regarding your concern about I/O overhead when loading 3D objects during online rendering, we have proactively addressed this issue by preprocessing the data, which significantly accelerates the loading process and effectively mitigates the I/O bottleneck. Furthermore, we plan to open-source all our code, giving you the opportunity to directly test our approach and verify the efficiency gains for yourself.
>
> We discuss the split-sum approximation extensively in our manuscript, as it not only enables online rendering, but also enables predictive rendering. Unlike previous LRMs that directly predict the final RGB color for rendering, we use the split-sum approximation for predictive rendering, which effectively models the specular component, thereby improving the reconstruction performance of specular surfaces. If the process is not clearly illustrated, other readers may not fully understand the rendering method. For instance, reviewer UZVs raised a question about the split-sum approximation and requested further clarification.
>
> Overall, we do not understand why the reviewer seems to disregard these other important contributions of our work.
>
> **And there is one point I would like to raise: How did you know that the Gradio demo was released by the first author of MeshLRM, as it is explicitly marked as an unofficial demo? I believe and hope that the review process for top conference ICLR can remain fair and unbiased, free from any malicious competition among peers.**

---

> > ### Comment · Reviewer_FrKV · 2024-12-02
> >
> > I strongly suggest the authors stay calm and maintain professionalism in their responses.  Venting anger and attacking reviewers does not benefit the review process at all.
> >
> >
> > First and foremost, I said the Gradio demo of MeshLRM is from the first author of MeshLRM simply because the demo was initially announced by her on her Twitter account, and additionally, she is affiliated with the company that released the demo (Hillbot), which made me reasonably assume she finished the demo. I have no idea why the authors say my negative reviews are **malicious competition among peers** just because I said the MeshLRM's demo was from its first author.  Leaving negative reviews because of peer competition is severe academic misconduct.  Accusing a reviewer of engaging in doing this without a factual basis is highly inappropriate and unwarranted.  **I believe the authors should sincerely apologize for their baseless attack on me.** To clarify, I am not the author of MeshLRM, nor have I submitted any mesh reconstruction papers to ICLR this year. Thus, the notion of “peer competition” between myself and this paper is entirely unfounded.
> >
> >
> > For the answer to the first question, I originally thought the split-sum was just for the training data because in L250-252, the authors say the *"split-sum enables online data preparation"* and the split-sum is seldom mentioned in the latter parts. After reading some responses, it appears that the “split-sum” may also be used for rendering predicted images. However, the authors use the term "predictive rendering", which is a confusing expression. This is why I explicitly asked whether the split-sum is applied to both the predicted and ground-truth results. A simple “yes” or “no” answer would have sufficed to resolve this question clearly. Using the term "predictive rendering" repeatedly keeps me confused.  I don't know why the authors go extremely mad about that i hope they can clarify this question. In addition, I suggest the author revise the writing or terms to make it less confusing.
> >
> >
> > For the answer to the second question, before we start, I would like to remind the authors again that venting anger and attacking reviewers does not benefit the review process at all.
> > I think there are some misunderstandings about our conversation. To clarify: I wanted to discuss the necessity of rendering the training data online with a split-sum approximation in previous responses , and I was not 100% sure if you also used split-sum to render the predicted images before(as I discussed above). Additionally, after you finally confirmed that you used split-sum to render the predicted images,  I also agree that using split-sum to render the predicted results during training and compute the losses can improve the final results. Based on your experiment, I feel the key reason for your performance improvement is neither rendering the **training data** online nor rendering them with split-sum approximation, but modeling the PBR for the predicted results during training and accelerating the PBR for the **predicted results** with split-sum approximation. (If you render the data offline without split-sum but using Blender with higher rendering quality, the performance may be better, and rendering data with Blender is not as slow as you describe if your implementation is correct and well-optimized). If you agree with this, please revise the writing to better clarify this.
> >
> > I believe any further discussion won't be helpful before the reviewer can stay calm and maintain professionalism in their responses.

---

> > > ### Author Response · Authors · 2024-12-03
> > >
> > > Thank you for your response.
> > >
> > > I believe there may be some misunderstanding regarding the use of PBR for rendering predicted images. I initially thought you understood the concept and was simply repeating the same question.
> > >
> > > In our manuscript, we mentioned in the abstract: "Moreover, by employing an explicit mesh as our 3D representation, PRM ensures the application of differentiable PBR, which supports the utilization of multiple photometric supervisions and better models the specular color for high-quality geometry optimization." We also included an arrow labeled "Differentiable PBR" from the predicted mesh to the predicted rendered maps in Figure 2. In Equation 11, we further emphasized that the predicted mesh is rendered using PBR with equation.
> > >
> > > If there is still any confusion regarding whether PBR is used for rendering predicted images, based on the information provided above, we are happy to modify our manuscript to clarify this. Additionally, we would like to emphasize that the split-sum approximation is a widely used real-time PBR method, to prevent any potential misunderstandings.
> > >
> > > Regarding the discussion on why the reconstruction performance improves, the reason we present in our manuscript is not due to rendering the training data online using split-sum approximation or PBR for rendering predicted images (which indeed improves performance, but is not the main contribution of our work). In fact, we never stated that preparing training data online via split-sum approximation is the primary factor behind the performance improvement, it only makes data preparation more flexible and effcient.
> > >
> > > The key factor is that the rendered training data should be photometric stereo images. For each object, multi-view images are rendered under varied camera poses, materials, and lighting conditions, as illustrated in Figure 10 (the images in the first row show examples of input images). This approach contrasts with using fixed, simple lighting and diffuse materials. However, photometric stereo images often include specular appearances, particularly when the roughness is low and metallic properties are high.
> > >
> > > Building on previous per-scene optimization methods like NeRO, modeling specular appearances with PBR has been shown to improve reconstruction performance. This explains why we chose PBR for rendering predicted images and how it contributes to performance improvement. However, the most important factor remains the use of photometric stereo images for training.
> > >
> > > While it is true that we could use Blender to render these images, the process would be highly time-consuming, at least on our machine. Therefore, we chose to employ split-sum approximation for online rendering, which allows us to improve efficiency without sacrificing the diversity and quality of the training data. We will include a discussion on this choice in our manuscript. Thansk for your suggestion.
> > >
> > > At the same time, we feel it is important to address an aspect of the review process that has impacted our ability to engage fully with your feedback. Specifically, your responses came primarily during the final stages of the rebuttal phase, after a prolonged period without feedback earlier in the process. This left limited time for a balanced and thorough discussion of the concerns raised. We hope that in future interactions, feedback can be distributed more evenly throughout the process to allow for a more constructive dialogue.
> > >
> > > Based on all factors discussed above, we have the final question. But we want to emphasize that our intention was not to make personal accusations but rather to understand the context of your remarks regarding the demo. If our response came across as inappropriate, we regret any misunderstanding and assure you that our aim was solely to seek clarification.

---

### Official Review · Reviewer_noz1 · 2024-11-03

**Soundness:** 3
**Presentation:** 3
**Contribution:** 3
**Rating:** 6
**Confidence:** 3

**Summary:**

This paper introduced a new LRM method that improves the output quality. The method addressed the limitation of LRMs that they rely on simple lighting and material assumptions as input. The paper integrates photometric stereo principles into large reconstruction models. The key innovations are: (1) using varied materials and lighting conditions during training to improve detail reconstruction and robustness, (2) incorporating real-time rendering with split-sum approximation for flexible online image generation, and (3) utilizing explicit mesh representation with differentiable physically-based rendering (PBR) for better geometry optimization.

**Strengths:**

- Novel integration of photometric stereo principles into large reconstruction models
- Comprehensive ablation studies that validate each component's contribution
- Practical applications demonstrated through relighting and material editing capabilities
- Impressive handling of specular surfaces, which are traditionally challenging

**Weaknesses:**

- Limited discussion of computational overhead compared to simpler approaches
- The 50% probability threshold for material/lighting consistency seems arbitrary
- Results appear sensitive to multi-view diffusion model quality
- Some failure cases (e.g., with lacking depth information) could be analyzed more thoroughly

**Questions:**

- How does the computational cost of online rendering compare to traditional offline approaches?
- What is the rationale behind the 50% probability threshold for material/lighting consistency?
- Have you considered incorporating depth estimation to improve reconstruction quality for challenging cases?
- How does the method perform on real-world images with unknown lighting conditions?

---

> ### Author Response · Authors · 2024-11-21
>
> $\textbf{Question1}$
>
> Limited discussion of computational overhead compared to simpler approaches.
>
> $\textbf{Answer1}$
>
> Our network mirrors the structure of InstantMesh [1], and the computational overhead is similarly efficient. The online split-sum approximation approach takes only 0.008 seconds to render a single image, ensuring it does not introduce any additional computational burden.
>
> [1] Xu J, Cheng W, Gao Y, et al. Instantmesh: Efficient 3d mesh generation from a single image with sparse-view large reconstruction models[J]. arXiv preprint arXiv:2404.07191, 2024.
>
> $\textbf{Question2}$
>
> The 50% probability threshold for material/lighting consistency seems arbitrary.
>
> $\textbf{Answer2}$
>
> Due to the high training cost for LRM, we did not carefully fine-tune the probabilities. The motivation for setting the threshold is that, in most cases, we input images captured under the same material and lighting conditions for reconstruction. Therefore, it is essential for the model to effectively handle this scenario as well.
>
> $\textbf{Question3}$
>
> Results appear sensitive to multi-view diffusion model quality.
>
> $\textbf{Answer3}$
>
> Yes, this is a common issue for 3D generation models, which highlights the importance of training a robust multi-view diffusion model. However, it is important to note that this limitation is not specific to our proposed method. Our work focuses on improving the reconstruction quality of LRM, given the generated multi-view images or real-captured sparse multi-view images.
>
> $\textbf{Question4}$
>
> Some failure cases (e.g., with lacking depth information) could be analyzed more thoroughly.
>
> $\textbf{Answer4}$
>
> The lack of depth information can lead to failures in generating multi-view images using a pretrained multi-view diffusion model. However, it is important to clarify that this is not a limitation of our model.
> Our work focuses on improving the reconstruction quality of LRM, given either generated multi-view images from multi-view images or captured sparse multi-view images. That said, you present an intriguing idea.
>
> We tested an advanced single-image depth estimation model [1], and the results are shown in Figure 16 in the revised PDF. If correct depth information is incorporated into the multi-view diffusion model, it could potentially enhance the accuracy of the generated multi-view images. We have discussed this potential solution in our manuscript. Thank you for your suggestion. I believe you have raised a meaningful question for improving existing multi-view diffusion models.
>
> [1] Yang L, Kang B, Huang Z, et al. Depth Anything V2[J]. arXiv preprint arXiv:2406.09414, 2024.
>
>
> $\textbf{Question4}$
>
> How does the method perform on real-world images with unknown lighting conditions?
>
> $\textbf{Answer4}$
>
> Our method is robust to variations in materials and lighting, as it utilizes photometric stereo images for both input and supervision. Figure 13 illustrates several examples. The object in the second row is from DTU [1], and the objects in the last four rows are from NeRO [2]; all objects are real captures under unknown lighting conditions. Please note that ground-truth lighting was only used during training. However, during inference, our method requires no additional information beyond the RGB images.
>
> [1] Jensen R, Dahl A, Vogiatzis G, et al. Large scale multi-view stereopsis evaluation[C]//Proceedings of the IEEE conference on computer vision and pattern recognition. 2014: 406-413.
>
> [2] L iu Y, Wang P, Lin C, et al. Nero: Neural geometry and brdf reconstruction of reflective objects from multiview images[J]. ACM Transactions on Graphics (TOG), 2023, 42(4): 1-22.

---

> ### Author Response · Authors · 2024-11-25
>
> Dear Reviewer noz1:
>
> We sincerely appreciate the time and effort you have dedicated to reviewing our paper and for providing such thoughtful feedback! As the discussion period concludes in two days, we kindly ask if our rebuttal has addressed your concerns to your satisfaction. If there are any remaining points requiring clarification or further improvement, please do not hesitate to let us know—we are fully committed to addressing them promptly.
> Thank you once again for your invaluable contributions to improving our work!
>
> Warm regards, The Authors

---

### Official Review · Reviewer_UZVs · 2024-11-04

**Soundness:** 3
**Presentation:** 3
**Contribution:** 3
**Rating:** 8
**Confidence:** 3

**Summary:**

This submission addresses the multi-view photometric stereo task and proposes a large reconstruction model (LRM) specifically leveraging photometric stereo. Unlike previous LRM models, in this work the authors train the model on data with varying illumination, material, etc. and instead of the usual triplane representation, they leverage a traditional mesh as the output 3D representation. This allows them to incorporate physically based rendering priors into the training objective. The idea of split-sum approximation from the physically based rendering community is leveraged to speedup the training data setup.

The approach is able to generate 3D models with fine geometric detail, handle a very wide variety of shapes, materials, lighting. Both qualitative and quantitative results are very convincing. However this is large transformer based architecture and training costs are prohibitively high.

**Strengths:**

The experimental results reported in the paper are very convincing. The performance of the model is quite impressive both from the point of view of the reconstructed 3D geometry, the large range of objects, materials that can be handled as well as the ability to handle a fairly small number of multi-view images.

The quantitative results reported in Tables 1 and 2 on the GSO and Omni3D datasets are quite convincing. A large improvement can be seen compared to prior LRM based baselines.

The method builds on top of the InstantMesh (Xu et al. 2024) model in terms of architecture and the training procedure. However, the model is extended to solve the photometric stereo problem. Leveraging mesh representations for geometry allows additional priors to be incorporated such as depth maps and normal maps (which are available for meshes).

**Weaknesses:**

This is a well written paper but the technical details were difficult to follow in certain sections. The section describing the use of the split-sum approximation was difficult to understand because it was quite brief. I was also unable to appreciate to what extent this approximation is needed. Is this a standard solution that is used in real-time rendering nowadays and what assumptions or requirements does this method have, which could potential make it inapplicable.

**Questions:**

How were the hyperparameters tuned and to what extent do the authors think that they matter, or specifically which ones matter more than the others? Given the high training cost, it is infeasible to carefully tune so many hyperparameters.

I am curious to know what happens on scenes with real, non-trivial backgrounds? Can the proposed model be run on multi-view images of non segmented real objects with natural backgrounds? Will the model reconstruct the object as well as the background? or would it be expected to automatically ignore the backgrounds because of how it was trained.

**Details Of Ethics Concerns:**

No concerns.

---

> ### Author Response · Authors · 2024-11-21
>
> $\textbf{Question1}$
>
> Details and concerns about split-sum approximation.
>
> $\textbf{Answer1}$
>
> The split-sum approximation is a widely used method in real-time rendering, as introduced in [1]. This approach is also extensively applied in per-scene optimization methods [2, 3]. In our manuscript, we adopt this method for photometric stereo image rendering due to its efficiency and fast computation speed. Specifically, this method accelerates rendering by approximating the integral in PBR.
> We provide some rendered examples in Figure 9, demonstrating photorealistic rendering quality. Notably, this method is a general-purpose real-time rendering technique that does not rely on any special assumptions or requirements.
> For further details, we recommend referring to [1]. Due to the complexity and length of the formula derivation, we are unable to include all the details in our manuscript. But do not worry about this issue, all of our code will be released.
>
> [1] Karis B, Games E. Real shading in unreal engine 4[J]. Proc. Physically Based Shading Theory Practice, 2013, 4(3): 1.
> [2] Munkberg J, Hasselgren J, Shen T, et al. Extracting triangular 3d models, materials, and lighting from images[C]//Proceedings of the IEEE/CVF Conference on Computer Vision and Pattern Recognition. 2022: 8280-8290.
> [3] Liu Y, Wang P, Lin C, et al. Nero: Neural geometry and brdf reconstruction of reflective objects from multiview images[J]. ACM Transactions on Graphics (TOG), 2023, 42(4): 1-22.
>
> $\textbf{Question2}$
>
> How are the hyperparameters tuned?
>
> $\textbf{Answer2}$
>
> As you mentioned, training LRM involves high computational costs, making it impractical to carefully fine-tune the hyperparameters. For the weights of the losses, we follow the approach used in InstantMesh. For the newly added losses, we adjust their weights based on the magnitude of the loss values to maintain balance.
>
> $\textbf{Question3}$
>
> what happens on scenes with real, non-trivial backgrounds?
>
> $\textbf{Answer3}$
>
> Most existing LRMs assume white background images as input, which can be easily obtained using Segment Anything (SAM) [1]. We experimented with multi-view images that include natural backgrounds as input, and the results are shown in Figure 19 in the revised PDF. Interestingly, the background is also reconstructed. While our model is not designed to handle images with backgrounds, your suggestion presents an intriguing idea. If input images with backgrounds were included during training, it is possible that the model could learn to handle such scenarios as well.
>
> [1] Kirillov A, Mintun E, Ravi N, et al. Segment anything[C]//Proceedings of the IEEE/CVF International Conference on Computer Vision. 2023: 4015-4026.

---

> > ### Comment · Reviewer_UZVs · 2024-11-26
> >
> > Dear authors, thanks for the quick response and the clarifications which addresses my main concerns.

---

### Official Review · Reviewer_Y5in · 2024-11-07

**Soundness:** 3
**Presentation:** 2
**Contribution:** 3
**Rating:** 6
**Confidence:** 4

**Summary:**

The paper proposes a variation of the large reconstruction models for reconstructing objects (geometry and albedo) from single or multi-view input images. By incorporating physically based rendering pipeline for image synthesis in training, as well as ground truth images rendered under sampled BRDF and lighting conditions, the model is able to utilize additional supervision based on diffuse and specular color maps, in an attempt to improve the generalization ability of the model to images of diverse and complex materials and lighting combinations. The model is evaluated against baselines include InstantMesh, and additional ablation is provided on losses related to PBR, as well as robustness to materials, number of views and field of view.

**Strengths:**

[1] The paper is able to improve on a line of work around LRMs, by incorporating PBR related insights into training, and showcases by incorporating ground truth and supervision from sampled complex materials and lighting conditions, the model better handles images of those conditions, and additionally gains improvement on geometry estimation due to the improved modeling of physics.

[2] The speed up of PBR rendering with split sum approximation enables efficient on-the-fly view synthesis and ground truth generation in a large parameter space of materials and lighting. Mostly as a technical contribution, but it will enable efficient data generation and augmentation on-the-fly when modeling complex parameter space of PBR.

[3] Extensive evaluation. The model is able to compare on standard benchmarks against baseline models in this task, but additionally provides extensive ablation study to justify the design choices by ablating the PBR-related losses, as well as robustness to number of input views and FOVs.

**Weaknesses:**

[1] Clarification. Several details need to be clarified to better understand the model and the training strategy.

(a) What does the model estimate w.r.t. the PBR parameters? Does it only estimate albedo?

(b) With sampled metallic, roughness and lighting envmaps, do we apply the metallic and roughness globally? If yes: 1)what happens if the original CAD model is already associated with spatially-varying (SV) BRDF maps? 2) And if applied globally, does this strategy diminish the model's generation ability towards real images with complex SV materials? 3) Given a good portion of Objaverse models are assigned with PBR materials, does it benefit the training to also predict ground truth BRDF (roughness, metallic) without manually sampling and enforcing global roughness and metallic?

(c) Is the split-sum approximation only applied to synthesizing estimated image from estimated representations, or it is also used to render ground truth images? Are ground truth images rendered on-the-fly for each batch in training?

[2] Writing. Language issues are abundant and need to be fixed for a polished version. Examples:

(a)  L015: for what purposes?

(b) L020: Need to introduce the full name of PBR before first use of the abbreviation.

(c) L050, L235: Need to clarify 'dependence on images rendered under fixed and simple lighting conditions' of previous methods. Mostly previous methods use PBR materials and envmap base lighting similar to this paper, so it would be important to clarify this assertion.

(d) L186: functionalities -> downstream applications of ...

(e) L283: what is 'a richer set of equations'?

[3] Additional evaluation results on images of complex lighting and materials. The paper is able to showcase the robustness towards complex lighting and materials in Fig. 9, however one scene is too few, and comparison with baselines on this setting is necessary to further justify the claim.

**Questions:**

Please see Weakness section for comments and questions.

---

> ### Author Response · Authors · 2024-11-21
>
> $\textbf{Question1}$
>
> What does the model estimate w.r.t. the PBR parameters? Does it only estimate albedo?
>
> $\textbf{Answer1}$
>
> Yes, we apply an albedo MLP for albedo prediction using the interpolated features from the triplane, similar to how previous LMRs perform RGB prediction. This is mentioned in Lines 308 and 733 of the manuscript.
> Predicting albedo instead of RGB is more reasonable because the interpolated triplane features represent positional information without accounting for view-dependent appearance. Albedo is a global, inherent, and view-independent property, making it a more suitable choice in this context. We have added a figure to illustrate the detailed network architecture in Figure 9 in the revised PDF.
>
> $\textbf{Question2}$
>
> Do we apply the metallic and roughness globally?
>
> what happens if the original CAD model is already associated with spatially-varying BRDF maps?
>
> $\textbf{Answer2}$
>
> Yes, we apply metallic and roughness properties globally. This approach stems from the challenge of assigning spatially-varying BRDFs, as we lack precise information on which parts of the object should be assigned specific BRDF values.
> For 3D models with spatially-varying BRDF maps, we replace these maps with our global values.
>
> $\textbf{Question3}$
>
> Does this strategy diminish the model's generation ability towards real images with complex SV materials?
>
> $\textbf{Answer3}$
>
> Through experiments, we found that this strategy does not compromise the model's ability to generate objects with complex spatially-varying materials. We provide several examples in Figure 18 in the revised PDF, showcasing results on both synthetic and real-captured images.
>
> $\textbf{Question4}$
>
> Does it benefit the training to also predict ground truth BRDF  without manually sampling and enforcing global roughness and metallic?
>
> $\textbf{Answer4}$
>
> Most objects in Objaverse are characterized by low metallicity and high roughness, which causes the diffuse term to dominate their appearance. If we do not adjust the metallic and roughness properties, specular surfaces would fall outside the domain of our model. We conducted an ablation study, as shown in Figure 8 to illustrate the importance of varying metallic and roughness.
>
> Predicting ground truth BRDF properties, such as roughness and metallic values, falls within the domain of inverse rendering. This task is typically more challenging, as demonstrated by previous per-scene optimization methods [1, 2].
> Incorporating these additional predictions would potentially increase the complexity of optimization, as a single triplane would need to simultaneously account for geometry, albedo, metallic, and roughness. Previous works [3] has proven that disentangled triplane can better model the textured 3D model.
>
> In this work, our primary focus is on improving the geometry reconstruction performance of LRMs. However, we are considering extending this research to develop an LRM framework for inverse rendering in the future. Achieving this would require a detailed understanding of each part of the object to assign appropriate materials to different regions. We sincerely thank you for inspiring us.
>
> [1] Zhang Y, Sun J, He X, et al. Modeling indirect illumination for inverse rendering[C]//Proceedings of the IEEE/CVF Conference on Computer Vision and Pattern Recognition. 2022: 18643-18652.
> [2] Jin H, Liu I, Xu P, et al. Tensoir: Tensorial inverse rendering[C]//Proceedings of the IEEE/CVF Conference on Computer Vision and Pattern Recognition. 2023: 165-174.
> [3] Liu Q, Zhang Y, Bai S, et al. DIRECT-3D: Learning Direct Text-to-3D Generation on Massive Noisy 3D Data[C]//Proceedings of the IEEE/CVF Conference on Computer Vision and Pattern Recognition. 2024: 6881-6891.
>
> $\textbf{Question5}$
>
> Is the split-sum approximation only applied to synthesizing estimated image from estimated representations, or it is also used to render ground truth images? Are ground truth images rendered on-the-fly for each batch in training?
>
> $\textbf{Answer5}$
>
> Yes, we also used the split-sum approximation for ground truth image rendering during training in an on-the-fly manner. Using Blender for ground truth rendering would be computationally prohibitive. For each 3D model, there are infinite possible camera poses, 121 material combinations, and 600 environment maps. Rendering all these combinations with Blender would take several months. We depicted this in Line 80- Line 82.

---

> ### Author Response · Authors · 2024-11-21
>
> $\textbf{Question6}$
>
> L015: for what purposes?
>
> $\textbf{Answer6}$
>
> We mentioned this in the last sentence: "Unlike previous large reconstruction models that prepare images under fixed and simple lighting as both input and supervision." The purpose here is to highlight the use of images as both input and supervision.
>
> $\textbf{Question7}$
>
> Need to introduce the full name of PBR before first use of the abbreviation.
>
> $\textbf{Answer7}$
>
> We have modified this in the revised version.
>
> $\textbf{Question8}$
>
> Need to clarify 'dependence on images rendered under fixed and simple lighting conditions' of previous methods.
>
> $\textbf{Answer8}$
>
> Most previous methods render images using only fixed, pre-set area or point lighting (simple lighting), with consistent lighting across all rendered images for each object. Examples of such methods include OpenLRM [1], SyncDreamer [2], and InstantMesh [3]. Notably, OpenLRM and SyncDreamer have provided their rendering scripts in their GitHub repositories for checking.
> In contrast, we render photometric images for both input and supervision. This means that each 3D model are rendered under varying lighting conditions for each view. Such images enhance the precision of local details by providing rich photometric cues and increase the model’s robustness to variations in the appearance of input images. This is also the key motivation of our work.
>
> [1] He, Zexin, and Tengfei Wang. "Openlrm: Open-source large reconstruction models." 2023,
> [2] Liu Y, Lin C, Zeng Z, et al. Syncdreamer: Generating multiview-consistent images from a single-view image[J]. arXiv preprint arXiv:2309.03453, 2023.
> [3] Xu J, Cheng W, Gao Y, et al. Instantmesh: Efficient 3d mesh generation from a single image with sparse-view large reconstruction models[J]. arXiv preprint arXiv:2404.07191, 2024.
>
> $\textbf{Question9}$
>
> what is 'a richer set of equations'?
>
> $\textbf{Answer9}$
>
> In traditional photometric stereo methods, each lighting condition corresponds to a specific rendering equation, as shown in Eq. (6). Observing the object under multiple lighting conditions provides several such equations, which imposes stronger constraints for surface normal recovery.
>
> $\textbf{Question10}$
>
> Additional evaluation results on images of complex lighting and materials.
>
> $\textbf{Answer10}$
>
> In Figure 9 (original version), we present rendered examples generated by our framework, which are not comparisons under complex lighting and materials.
>
> In fact, we show such cases in Figure 13. In the revised version, we have added comparisons on these objects with InstantMesh, the state-of-the-art baseline.
> Additionally, as noted in Table 1 and Table 2, we have rendered images under complex lighting for qualitative comparisons, as described in Line 367. Moreover, we have included qualitative results on selected objects including both synthetic and real-captured objects with spatially varying materials, as shown in Figure 18.

---

> > ### Comment · Reviewer_Y5in · 2024-11-23
> >
> > Thanks to the authors for providing additional results to highlight the improved geometry as a result of the PBR training strategy, in Fig. 13 of the revised manuscript. However it would be important to also provide ablation study by turning off each components. For instance, results of (1) switch to point light but keep the sampled metallic and roughness (2) keep image-based lighting, but use fixed metallic or roughness levels, or use original metallic and roughness from Objaverse assets. Those ablation study is essential to understand the contribution of each components to the improved final results.
> >
> > On the writing issues, I am able to understand most if not all of the text even if the language, vocabulary or grammar sometimes feels off. I would recommend additional proofreading using language tools or large language models to improve the writing for better clarification

---

> > > ### Author Response · Authors · 2024-11-23
> > >
> > > Thank you for your comments.
> > >
> > > Regarding the ablation study, we have conducted an experiment using the fixed, original metallicity and roughness values from Objaverse (see Line 518). The qualitative comparison is provided in Figure 8. Additionally, we have performed a quantitative comparison in the revised version, and the results are reported in the table below.
> > >
> > > | **Metric**               | **CD↓** | **FS @ 0.1↑** | **PSNR↑** | **SSIM↑** | **LPIPS↓** |
> > > |---------------------------|----------|--------------|-----------|-----------|------------|
> > > | w/o change materials      | 0.089    | 0.894        | 19.662    | 0.817     | 0.159      |
> > > | Full model                | **0.066**    | **0.948**        | **20.992**    | **0.830**     | **0.137**      |
> > >
> > > As for the lighting, photometric stereo images need to be rendered under varying lighting conditions. The variation provided by point lighting is far less significant than that achieved using image-based lighting. Moreover, image-based lighting introduces more high-frequency specular components, which are beneficial for recovering fine-grained local details. We believe that image-based lighting is a better choice for rendering photometric stereo images. However, I am happy to conduct an ablation study to validate this point in the revised version.

---

> > > > ### Comment · Reviewer_Y5in · 2024-11-23
> > > >
> > > > Thanks to the authors for the swift response. Yes, the quality and amount of SV PBR materials and assignments in Objaverse is an issue, and extending the model to full inverse rendering will face challenges such as ambiguity. It might be possible to use the subset of Objaverse assets with PBR materials for training, or find a way to augment the materials. I would encourage the authors to add those discussion and ablation in the revised version. But overall I will stick to the original rating of weak AC.

---

> > > > > ### Author Response · Authors · 2024-11-23
> > > > >
> > > > > Thanks for the reviewer’s constructive suggestions. We will discuss these points in our revised manuscript and explore augmenting SV materials to extend our work into an inverse rendering framework in the future.

---

> ### Comment · Reviewer_Y5in · 2024-11-23
>
> Thanks to the authors for the response. There are a few further clarifications and questions that I would like to raise:
>
> Follow up to Question 1: just to confirm: does the model estimate roughness and metallic? In other words, are the estimated diffuse and specular color maps rendered with GT roughness and metallic, or estimated ones? The revised Fig. 9 shows only albedo is estimated. What are the obstacles to also estimated roughness and metallic, and go a step further to estimated SV roughness and metallic?
>
> Acknowledging the response from the authors, SV metallic and roughness are not estimated, and as a result the model only produce a 3D mesh with diffuse textures only? In this the case, it will inherently **limit applications** such as relighting where the original roughness and metallic are not captured, as a result specularities cannot be reproduced. It would help clarify the contributions by explicitly stating that the method is only focused on geometry and albedo, and adding additional discussion on this limitation and how the model can possibly be extended for full inverse rendering.

---

> > ### Author Response · Authors · 2024-11-23
> >
> > Thank you for your comments.
> >
> > Our model does not estimate roughness or metallicity. During predictive rendering, we use the ground truth metallicity and roughness values, consistent with those used in ground-truth rendering. Estimating metallicity and roughness falls within the scope of inverse rendering, which is typically more challenging due to ambiguity issues, as different material properties can produce identical rendering results. This challenge has been widely documented in previous per-scene optimization inverse rendering frameworks.
> >
> > Further estimating spatially-varying (SV) roughness and metallicity would require sufficient SV material annotations. While some objects in Objaverse do feature SV materials, each model typically has only a single material annotation. Without manually assigning SV materials, these annotations are insufficient. The critical challenge lies in determining how to assign SV materials to an object, which requires an understanding of the object's parts to make assignments.
> >
> > We will emphasize in our revised manuscript that our model focuses on geometry reconstruction and discuss potential solutions for extending our work to inverse rendering.

---

### Author Response · Authors · 2024-11-21

We would like to thank all the reviewers for their valuable comments and feedback. We appreciate that the reviewers recognize the novelty of our motivation in using photometric stereo images for LRM and acknowledge the significance of our results.
We have revised our paper based on the reviewers' comments, with changes highlighted in red in the updated PDF. Before addressing the reviewers’ feedback in detail, point by point, we kindly request that the reviewers and the Area Chair review this blog content first. It primarily addresses potential misunderstandings and common concerns regarding our paper. For specific questions, we will provide detailed responses in each reviewer's section.

$\textbf{1.Regarding the key motivation of our work.}$

Our primary motivation is to demonstrate that multi-view photometric stereo images can not only enhance precise local details but also improve the model's robustness to variations in the appearance of input images for LRM. Multi-view photometric stereo involves varying the material, lighting map, and camera poses when rendering each view of an object. This approach differs from previous methods, which render each view using simple, fixed area or point lighting and Lambertian materials.
Split-sum approximation serves as a tool to enable online rendering, offering greater flexibility for ground-truth rendering. For predictive rendering, split-sum approximation is essentially a PBR method that excels at modeling the specular component. This capability facilitates improved reconstruction performance on glossy surfaces, as demonstrated in prior per-scene optimization methods such as NeRO [1].

[1] Liu Y, Wang P, Lin C, et al. Nero: Neural geometry and brdf reconstruction of reflective objects from multiview images[J]. ACM Transactions on Graphics (TOG), 2023, 42(4): 1-22.

$\textbf{2.Regarding the computational cost of online rendering compared to traditional offline approaches.}$

The split-sum approximation approach can render photorealistic images at significantly faster speeds compared to traditional offline methods with Blender. Specifically, split-sum approximation renders an image in just 0.008 seconds, whereas traditional offline methods take over 1 second per image.

---

### Meta-Review · Area_Chair_5dX1 · 2024-12-20

**Metareview:**

This paper presents a photometric stereo-based large reconstruction model to reconstruct high-quality mesh with fine-grained local details from multiview photometric stereo images. The strength of the work were the introduction introduction of PBR-related insights into the training of large reconstruction models, the performance is quite impressive over the existing LRMs. The downsides were that the technical contribution of the work was rather limited due to that the use of richer data (multiview photometric stereo images) to improve the performance was quite straightforward. There was an extensive discussion at the reviewer-author discussion phase, and the reviewers' final ratings were mixed, putting the paper at the borderline. During the reviewer-AC discussion phase, none of the reviewers was excited about the paper regardless of their paper ratings. The AC weighed more on the limited novelty aspect of the work reached this decision.

**Additional Comments On Reviewer Discussion:**

During the reviewer-author discussion phase, there was an extensive discussion about the need for online rendering, which unfortunately did not converge. This point was not critical in making the decision. The use of multiview photometric stereo images in LRM for improving the reconstruction accuracy is rather straightforward as it is understood that the richer input and training data improve the estimation accuracy. Other than this usage of the multiview photometric stereo images, the paper's technical contributions were rather limited.

---

### Decision · Program_Chairs · 2025-01-22

Reject